# Is Your HD Map Constructor Reliable under Sensor Corruptions?

**Xiaoshuai Hao**[1]    **Mengchuan Wei**[1]    **Yifan Yang**[1]    **Haimei Zhao**[2]
**Hui Zhang**[1]    **Yi Zhou**[1]    **Qiang Wang**[1]    **Weiming Li**[1]
**Lingdong Kong**[3,✉]    **Jing Zhang**[2,✉]

[1]Samsung R&D Institute China–Beijing
[2]The University of Sydney    [3]National University of Singapore
{xshuai.hao,mc.wei,yifan.yang,hui123.zhang,yi0813.zhou}@samsung.com
{qiang.w,weiming.li}@samsung.com   hzha7798@uni.sydney.edu.au
lingdong@comp.nus.edu.sg   jing.zhang1@sydney.edu.au

https://mapbench.github.io

## Abstract

Driving systems often rely on high-definition (HD) maps for precise environmental information, which is crucial for planning and navigation. While current HD map constructors perform well under ideal conditions, their resilience to real-world challenges, *e.g.*, adverse weather and sensor failures, is not well understood, raising safety concerns. This work introduces MapBench, the first comprehensive benchmark designed to evaluate the robustness of HD map construction methods against various sensor corruptions. Our benchmark encompasses a total of 29 types of corruptions that occur from cameras and LiDAR sensors. Extensive evaluations across 31 HD map constructors reveal significant performance degradation of existing methods under adverse weather conditions and sensor failures, underscoring critical safety concerns. We identify effective strategies for enhancing robustness, including innovative approaches that leverage multi-modal fusion, advanced data augmentation, and architectural techniques. These insights provide a pathway for developing more reliable HD map construction methods, which are essential for the advancement of autonomous driving technology. The benchmark toolkit and affiliated code and model checkpoints have been made publicly accessible.

## 1 Introduction

HD maps are fundamental components in autonomous driving systems, providing centimeter-level details of traffic rules, vectorized topology, and navigation information [35, 43]. These maps enable the ego-vehicle to accurately locate itself on the road and anticipate upcoming features [41, 74, 11, 50]. HD map constructors formulate this task as predicting a collection of vectorized static map elements in bird's eye view (BEV), *e.g.*, pedestrian crossings, lane dividers, road boundaries, *etc.* [42, 73, 77].

Existing HD map construction methods can be categorized based on the input sensor modality: camera-only [35, 43, 41, 74, 50] LiDAR-only [35, 43, 41, 42], and camera-LiDAR fusion [35, 43, 41] models. Each sensor poses distinct functionalities: cameras capture semantic-rich information from images, while LiDAR provides explicit geometric information from point clouds [42, 48, 26, 29]. Generally, camera-based methods outperform LiDAR-only methods, and fusion-based methods yield the most satisfactory results [31, 77]. However, current model designs and performance evaluations are based on ideal driving conditions, *e.g.*, clear daytime weather and fully functional sensors [3, 34].

---

✉ Lingdong Kong and Jing Zhang are the corresponding authors of this work.

38th Conference on Neural Information Processing Systems (NeurIPS 2024) Track on Datasets and Benchmarks.

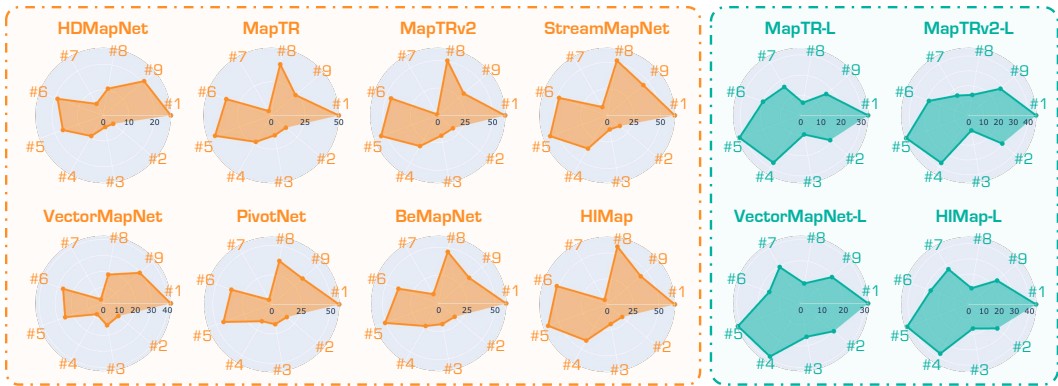

Figure 1: Radar charts of state-of-the-art HD map constructors under the Camera and LiDAR sensor corruptions. We report the mAP scores of different map construction methods under each corruption type across severity levels. **Camera Corruptions:** #1 Clean, #2 Frame Lost, #3 Camera Crash, #4 Low-Light, #5 Bright, #6 Color Quant, #7 Snow, #8 Fog, and #9 Motion Blur. **LiDAR Corruptions:** #1 Clean, #2 Wet Ground, #3 Snow, #4 Motion Blur, #5 Incomplete Echo, #6 Fog, #7 Crosstalk, #8 Cross-Sensor, and #9 Beam Missing. The radius of each chart is normalized based on the Clean score. The larger the area coverage, the better the overall robustness.

In real-world driving scenarios, adverse conditions, such as bad weather, motion distortions, and sensor malfunctions (frame loss, sensor crashes, incomplete echoes, *etc.*) are unavoidable [62, 27]. It remains unclear how existing HD map construction methods perform under such challenging yet safety-critical conditions, highlighting the need for a thorough out-of-domain robustness evaluation.

To address this gap, we introduce MapBench, the first comprehensive benchmark aimed at evaluating the reliability of HD map construction methods against natural corruptions that occur in real-world environments. We thoroughly assess the model's robustness under corruptions by investigating three popular configurations: camera-only, LiDAR-only, and camera-LiDAR fusion models. Our evaluation encompasses 8 types of camera corruptions, 8 types of LiDAR corruptions, and 13 types of camera-LiDAR corruption combinations, as depicted in Fig. 2 and Fig. 4. We define three severity levels for each corruption type and devise appropriate metrics for quantitative robustness comparisons.

Utilizing MapBench, we perform extensive experiments on a total of 31 state-of-the-art HD map construction methods. The results, as shown in Fig. 1, reveal significant discrepancies in model performance across "clean" and corrupted datasets. Key findings from these evaluations include:

1) Among all camera/LiDAR corruptions, Snow corruption significantly degrades model performance; it covers the road, rendering map elements unrecognizable and posing a major threat to autonomous driving. Besides, sensor failure corruptions (*e.g.*, Frame Lost and Incomplete Echo) are also challenging for all models, demonstrating the serious threats of sensor failures on HD map models.

2) While Camera-LiDAR fusion methods have shown promising performance by incorporating information from both modalities [41, 77], existing methods often assume access to complete sensor information, leading to poor robustness and potential collapse when sensors are corrupted or missing.

Through extensive benchmark studies, we further unveil crucial factors for enhancing the reliability of HD map constructors against sensor corruption. The key contributions of this work are three-fold:

- We introduce MapBench, making the first attempt to comprehensively benchmark and evaluate the robustness of HD map construction models against various sensor corruptions.

- We extensively benchmark a total of 31 state-of-the-art HD map constructors and their variants under three configurations: camera-only, LiDAR-only, and camera-LiDAR fusion. This involves studying their robustness to 8 types of camera corruptions, 8 types of LiDAR corruptions, and 13 types of camera-LiDAR corruption combinations for each configuration.

- We identify effective strategies for enhancing robustness, including innovative approaches that leverage advanced data augmentation and architectural techniques. Our findings reveal strategies that significantly improve performance and robustness, underscoring the importance of tailored solutions to address specific challenges in HD map construction.

## 2   Related Work

**HD Map Construction.** The construction of HD maps is a critical yet extensively researched area. Based on the input sensor modality, existing literature can be categorized into camera-only [41, 74, 11, 50, 42, 73, 77, 15], LiDAR-only [35, 43], and camera-LiDAR fusion [41, 42, 77] models. *Camera-based methods* [35, 43, 41, 74, 11, 50, 42, 73] have increasingly employed the BEV representation as an ideal feature space for multi-view perception due to its ability to mitigate scale ambiguity and occlusion challenges. Techniques such as LSS [49], Deformable Attention [39], and GKT [5] have been proposed to project perspective view (PV) features into the BEV space by leveraging geometric priors. However, these methods lack explicit depth information. Consequently, they have come to rely on higher resolution images or larger backbones to achieve enhanced accuracy [45, 44, 58, 39, 69, 65]. *LiDAR-based methods* [56, 35, 43, 42, 41, 38] benefit from the accurate 3D geometric information provided by LiDAR inputs but face challenges related to data sparsity and sensing noise.

**Multi-Sensor HD Map Construction.** *Camera-LiDAR fusion-based methods* can be roughly divided into three categories: point-level fusion [54, 53, 66, 8, 70], feature-level fusion [71, 1, 6, 7, 40], and BEV-level fusion [41, 42, 77, 17]. Recently, camera-LiDAR feature fusion in the unified BEV space has gained attention. BEV-level fusion integrates features from camera and LiDAR sensors within the same BEV space, combining complementary modalities to achieve superior performance over uni-modal approaches. Despite the progress in HD map construction methods, their resilience to real-world challenges such as adverse weather and sensor failures remains unclear, raising safety concerns [28, 30]. In this work, we make the first attempt to explore the robustness of existing HD map construction methods under sensor corruptions that occur in real-world environments.

**Robustness against Sensor Corruptions.** Assessing the robustness of driving perception models under sensor corruptions has emerged as a crucial research area [16, 12, 78, 28, 62, 30]. Recently, the corruption robustness of BEV perception tasks has been extensively studied. RoboDepth [28, 51] establishes a robustness benchmark for monocular depth estimation under corruptions, while RoboBEV [62, 63] introduces a comprehensive benchmark for evaluating the robustness of four BEV perception tasks, including 3D object detection [39, 46], semantic segmentation [75, 76], depth estimation [59], and semantic occupancy prediction [60, 24]. However, RoboBEV's analyses of multi-modal fusion model robustness only consider complete sensor failure, overlooking other common sensor corruptions and their combinations. Dong *et al.* [12] systematically design 27 types of common corruptions for 3D object detection in both LiDAR and camera sensors. Meanwhile, Robo3D [27] benchmarks the robustness of 3D detectors and segmentors against LiDAR corruptions.

**Comparison with Existing Works.** This work differs from prior literature in *three* key aspects. Firstly, we focus on the vectorized HD map construction task, distinct from other BEV perception tasks [12, 28, 62]. Secondly, we introduce new sensor corruption types that closely mimic real-world scenarios. Specifically, we design 13 new multi-sensor corruption types to benchmark camera-LiDAR fusion models comprehensively, surpassing the scope of complete sensor failure analysis in RoboBEV [62]. Thirdly, we explore distinct data augmentation techniques that are applied to LiDAR point clouds and RGB images to analyze their impact on enhancing corruption robustness. To the best of our knowledge, MapBench serves as the first study to comprehensively benchmark and evaluate the reliability of HD map construction methods again single- and multi-modal sensor corruptions.

## 3   MapBench: Benchmarking HD Map Construction Robustness

In this work, we investigate three popular configurations, *i.e.*, Camera-only, LiDAR-only, and Camera-LiDAR fusion-based HD map construction tasks, and study their robustness to various sensor corruptions. As illustrated in Fig. 2, the camera/LiDAR corruptions are grouped into exterior environments, interior sensors, and sensor failure types, covering the majority of real-world cases.

Following the protocol established in [20], we consider *three* corruption severity levels, *i.e.*, Easy, Moderate, and Hard, for each type of corruption. Additionally, regarding multi-sensor corruptions, we use camera/LiDAR sensor failure types to perturb camera and LiDAR sensor inputs separately or concurrently. MapBench is constructed by corrupting the *val* set of nuScenes [3]. We chose nuScenes since it has been widely utilized among almost all recent HD map construction works.

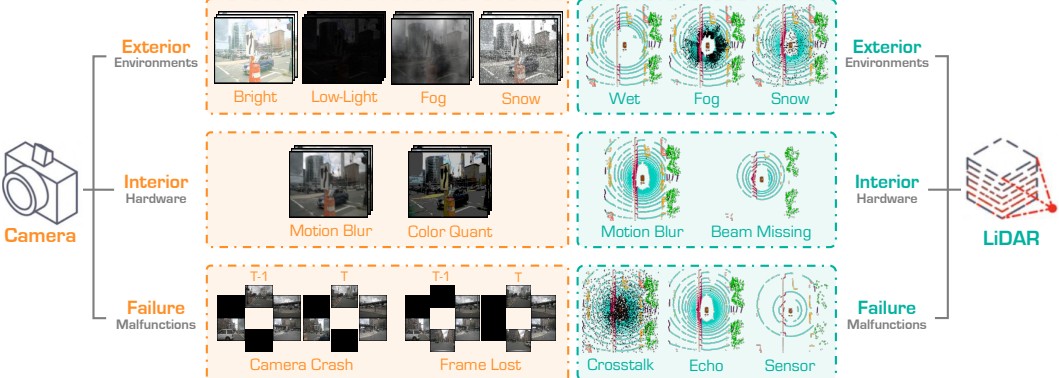

Figure 2: Definitions of the Camera and LiDAR sensor corruptions in MapBench. Our benchmark encompasses a total of 16 corruption types for HD map construction, which can be categorized into exterior, interior, and sensor failure scenarios. Besides, we define 13 multi-sensor corruptions by combining the camera and LiDAR sensor failure types. Kindly refer to our Appendix for more details.

## 3.1 Sensor Corruptions

**Camera Sensor Corruptions.** To probe the Camera-only model robustness, we employ 8 real-world camera sensor corruptions from [62], ranging from three perspectives: exterior environments, interior sensors, and sensor failures. Specifically, the exterior environments include various lighting and weather conditions such as `Bright`, `Low-Light`, `Fog`, and `Snow`. The camera inputs might also be corrupted by interior factors caused by sensors, such as `Motion Blur` and `Color Quantization`. Lastly, we consider the sensor failure cases where cameras crash or certain frames are dropped due to physical problems, leading to `Camera Crash` and `Frame Lost`, respectively. Due to page limits, the detailed definitions and visualizations of these corruptions are provided in Sec. A in the Appendix.

**LiDAR Sensor Corruptions.** To explore the LiDAR-only model robustness, we resort to 8 LiDAR sensor corruptions in [27], which are scenarios that have a high likelihood of occurring in real-world deployments. These corruptions also range from exterior, interior, and sensor failure cases. The exterior environments encompass `Fog`, `Wet Ground`, and `Snow`, which cause back-scattering, attenuation, and reflection of the LiDAR pulses. Besides, the LiDAR inputs might be corrupted by bumpy surfaces, dust, or insects, which often lead to disturbances and cause `Motion Blur` and `Beam Missing`. Lastly, we consider the cases of LiDAR internal sensor failures, such as `Crosstalk`, possible `Incomplete Echo`, and `Cross-Sensor` scenarios. Kindly refer to Sec. A for more details.

**Multi-Sensor Corruptions.** To explore the Camera-LiDAR fusion model robustness, we design 13 types of camera-LiDAR corruption combinations that perturb both camera and LiDAR input separately or concurrently, using the aforementioned sensor failure types. These multi-sensor corruptions are grouped into camera-only corruptions, LiDAR-only corruptions, and their combinations, covering the majority of real-world scenarios. Specifically, we design 3 camera-only corruptions by utilizing the "clean" LiDAR point data and three camera failure cases such as `Unavailable Camera` (all pixel values are set to *zero* for all RGB images), `Camera Crash`, and `Frame Lost`. Moreover, we design 4 LiDAR-only corruptions by utilizing the "clean" camera data and the corrupted LiDAR data as the input. This includes complete LiDAR failure (since no model can work when all points are absent, we approximate this scenario by only retaining a single point as input), `Incomplete Echo`, `Crosstalk`, and `Cross-Sensor`. Note that our implementations of complete LiDAR failure are close to the real-world situation. Lastly, we design 6 camera-LiDAR corruption combinations that perturb both sensor inputs concurrently, using the previously mentioned image/LiDAR sensor failure types. Due to page limits, more detailed definitions of multi-sensor corruption are placed in Sec. B.

## 3.2 Evaluation Metrics

Inspired by [20, 27, 62], we define two robustness evaluation metrics based on `mAP` (mean Average Precision), a commonly-used accuracy indicator for vectorized HD map construction.

Table 1: **Benchmarking HD map constructors.** Methods are split into groups based on [1]input modality, [2]BEV encoder, [3]backbone, and [4]training epochs. "L" and "C" represent LiDAR and camera, respectively. "Effi-B0", "R50", "PP", and "SEC" are short for EfficientNet-B0 [52], ResNet50 [18], PointPillars [32], and SECOND [68]. AP denotes performance on the clean nuScenes *val* set. The subscripts $b.$, $p.$, and $d.$ are short for the *boundary*, *pedestrian crossing*, and *divider*, respectively.

| Method | Venue | Modal | BEV Encoder | Backbone | Epoch | $AP_p.\uparrow$ | $AP_d.\uparrow$ | $AP_b.\uparrow$ | mAP↑ | mRR↑ | mCE↓ |
|---|---|---|---|---|---|---|---|---|---|---|---|
| HDMapNet [35] | ICRA'22 | C | NVT | Effi-B0 | 30 | 14.4 | 21.7 | 33.0 | 23.0 | 43.3 | 187.8 |
| VectorMapNet [43] | ICML'23 | C | IPM | R50 | 110 | 36.1 | 47.3 | 39.3 | 40.9 | 40.6 | 148.5 |
| PivotNet [11] | ICCV'23 | C | PersFormer | R50 | 30 | 53.8 | 58.8 | 59.6 | 57.4 | 45.2 | 96.3 |
| BeMapNet [50] | CVPR'23 | C | IPM-PE | R50 | 30 | 57.7 | 62.3 | 59.4 | 59.8 | 50.3 | 78.5 |
| MapTR [41] | ICLR'23 | C | GKT | R50 | 24 | 46.3 | 51.5 | 53.1 | 50.3 | 49.3 | 100.0 |
| MapTRv2 [42] | arXiv'23 | C | BEVPool | R50 | 24 | 59.8 | 62.4 | 62.4 | 61.5 | 51.4 | 72.6 |
| StreamMapNet [73] | WACV'24 | C | BEVFormer | R50 | 30 | 61.7 | 66.3 | 62.1 | 63.4 | 54.4 | 64.8 |
| HIMap [77] | CVPR'24 | C | BEVFormer | R50 | 24 | 62.2 | 66.5 | 67.9 | 65.5 | 56.6 | 56.9 |
| VectorMapNet [43] | ICML'23 | L | - | PP | 110 | 25.7 | 37.6 | 38.6 | 34.0 | 63.4 | 94.4 |
| MapTR [41] | ICLR'23 | L | - | SEC | 24 | 48.5 | 53.7 | 64.7 | 55.6 | 55.1 | 100.0 |
| MapTRv2 [42] | arXiv'23 | L | - | SEC | 24 | 56.6 | 58.1 | 69.8 | 61.5 | 57.2 | 74.6 |
| HIMap [77] | CVPR'24 | L | - | SEC | 24 | 54.8 | 64.7 | 73.5 | 64.3 | 59.2 | 73.1 |
| MapTR [41] | ICLR'23 | C & L | GKT | R50 & SEC | 24 | 55.9 | 62.3 | 69.3 | 62.5 | 57.1 | 100.0 |
| HIMap [77] | CVPR'24 | C & L | BEVFormer | R50 & SEC | 24 | 71.0 | 72.4 | 79.4 | 74.3 | 41.7 | 110.6 |

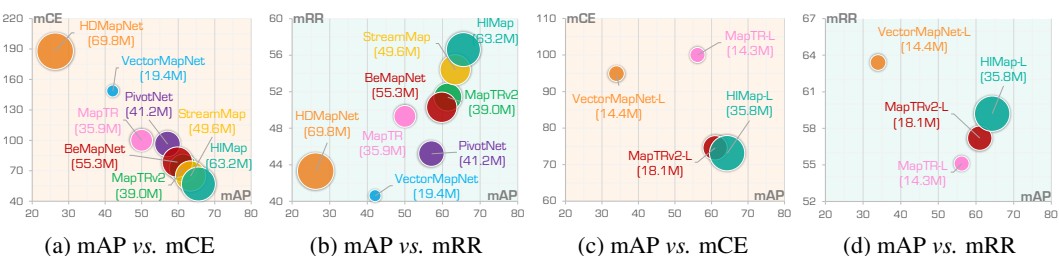

(a) mAP *vs.* mCE     (b) mAP *vs.* mRR     (c) mAP *vs.* mCE     (d) mAP *vs.* mRR

Figure 3: The correlations of accuracy (mAP) and robustness (mCE / mRR) for the Camera (a) and (b) and LiDAR (c) and (d) models. The size of the circle represents the number of model parameters.

**Corruption Error (CE).** We define CE as the primary metric in comparing models' robustness. It measures the relative robustness of candidate models compared to a baseline. Given a total of $N$ distinct corruption types, the CE and mCE (mean Corruption Error) scores are calculated as follows:

$$\text{CE}_i = \frac{\sum_{l=1}^{3}(1 - \text{mAP}_{i,l})}{\sum_{l=1}^{3}(1 - \text{mAP}_{i,l}^{\text{base}})}, \quad \text{mCE} = \frac{1}{N}\sum_{i=1}^{N}\text{CE}_i, \quad (1)$$

where $i$ denotes the corruption type and $l$ is the severity level. $\text{mAP}_{i,l}^{\text{base}}$ is the baseline's accuracy score.

**Resilience Rate (RR).** We define RR as the relative robustness indicator for measuring how much accuracy a model can retain when evaluated on the corruption sets, which are calculated as follows:

$$\text{RR}_i = \frac{\sum_{l=1}^{3}\text{mAP}_{i,l}}{3 \times \text{mAP}^{\text{clean}}}, \quad \text{mRR} = \frac{1}{N}\sum_{i=1}^{N}\text{RR}_i, \quad (2)$$

where $\text{mAP}^{\text{clean}}$ denotes the mAP score of a candidate model on the "clean" evaluation set.

## 4 Experimental Analysis

### 4.1 Benchmark Configuration

**Candidate Models.** Our MapBench encompasses a total of 31 HD map constructors and their variants, *i.e.*, HDMapNet [35], VectorMapNet [43], PivotNet [11], BeMapNet [50], MapTR [41], MapTRv2 [42], StreamMapNet [73] and HIMap [77]. The code of some other HD map methods [36, 64, 22, 74, 67, 47, 4, 25, 65, 72, 55] are not open source, thus will not be considered in this work.

**Model Configurations.** We report the basic information of different models in Tab. 1, including input modality, BEV encoder, backbone, training epoch, and their performance on the official nuScenes

Table 2: Ablation on the use of BEV encoders.

| Method | Encode | $AP_{p.}$ | $AP_{d.}$ | $AP_{b.}$ | mAP | mRR | mCE |
|---|---|---|---|---|---|---|---|
| MapTR ○ | BEVFormer | 43.7 | 49.8 | 52.6 | 48.7 | 49.3 | 100.0 |
| MapTR ○ | BEVPool | 44.9 | 51.9 | 53.5 | 50.1 | 48.1 | 99.3 |
| MapTR ● | GKT | 46.3 | 51.5 | 53.1 | 50.3 | 49.3 | 97.2 |

Table 3: Ablation on the use of temporal fusion.

| Method | Temp | $AP_{p.}$ | $AP_{d.}$ | $AP_{b.}$ | mAP | mRR | mCE |
|---|---|---|---|---|---|---|---|
| StreamMap ○ | ✗ | 17.2 | 22.6 | 31.6 | 23.8 | 47.1 | 100.0 |
| StreamMap ● | ✓ | **21.4** | **27.4** | **35.2** | **28.0** | **55.5** | **85.9** |

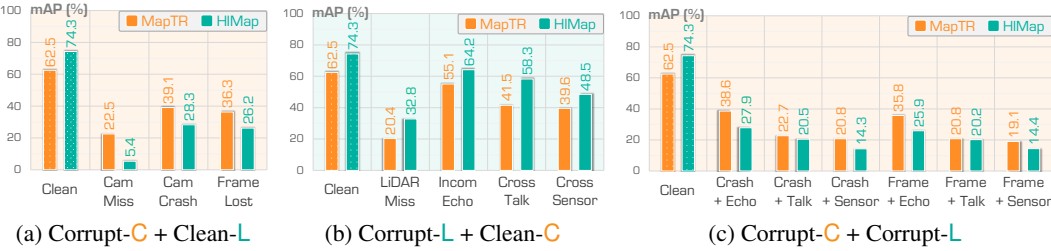

(a) Corrupt-C + Clean-L          (b) Corrupt-L + Clean-C          (c) Corrupt-C + Corrupt-L

Figure 4: The results of Camera-LiDAR fusion methods [41, 77] under multi-sensor corruptions.

validation set. Note that the LiDAR-only models here take temporally aggregated LiDAR points as the input, hence their mAP on "clean" data are much higher than those from other tables or figures, where single-scan LiDAR points are utilized for a fair comparison with the corrupted data.

**Evaluation Protocol.** To ensure fairness, we use official model configurations and public checkpoints provided by open-sourced codebase whenever applicable, or re-train the model following default settings. Furthermore, we report metrics for each corruption type by averaging over three severity levels. We adopt MapTR [41] under different configurations (see Tab. 1) as our baseline for calculating the mCE metric in Eq. 1, considering its wide adoption among state-of-the-art methods.

## 4.2 Camera-Only Benchmarking Results

We show the camera sensor corruption robustness of 8 camera-only HD map models in Fig. 3 (a)-(b). Our findings indicate that existing HD map models exhibit varying degrees of performance declines under corruption scenarios. Overall, the corruption robustness is highly correlated with the original accuracy on the "clean" data, as the models (*e.g.*, StreamMapNet [73], HIMap [77]) with higher accuracy also achieve better corruption robustness. We further show the accuracy comparisons of camera-only methods under different corruption severity levels in Fig. 6. Based on the empirical evaluation results, we draw several important findings, which can be summarized as follows:

**1)** We observe that among all camera corruptions, Snow degrades performance the most, which poses a significant threat to driving safety. The main reason is that Snow will cover the road, causing the map element to be unrecognizable. Besides, Frame Lost and Camera Crash are also challenging for all models, demonstrating the serious threats of camera sensor failures on camera-only models.

**2)** As shown in Fig. 3 (a)-(b), the two most robust models are StreamMapNet [73] and HIMap [77]. Although they achieve better robustness under various camera corruptions than other studied models, the overall robustness of existing models is still relatively low. Specifically, the mRR ranges from 40% to 60%, and the best HIMap [77] model only yields 56.6%. For more detailed experimental results in terms of class-wise CE and RR, kindly refer to Tab. 14 to Tab. 17 in Sec. D in the Appendix.

## 4.3 LiDAR-Only Benchmarking Results

We report the LiDAR sensor corruption robustness of 4 LiDAR-only HD map constructors in Fig. 3 (c)-(d) and Fig. 6. Similar to the observations of camera-only models, LiDAR-only models that have higher accuracy on the "clean" set generally achieve better corruption robustness. Key aspects are:

**1)** Among all corruptions, Snow and Cross-Sensor impair performance the most, posing a significant threat to the robustness of LiDAR-only methods. More specifically, both Snow and Cross-Sensor lead to more than 80% performance drops for all LiDAR-only methods. The main reason is that Snow causes laser pulse reflections in LiDAR data. Besides, Cross-Sensor shows that the domain gap caused by variant LiDAR configurations/devices reduces the performance greatly.

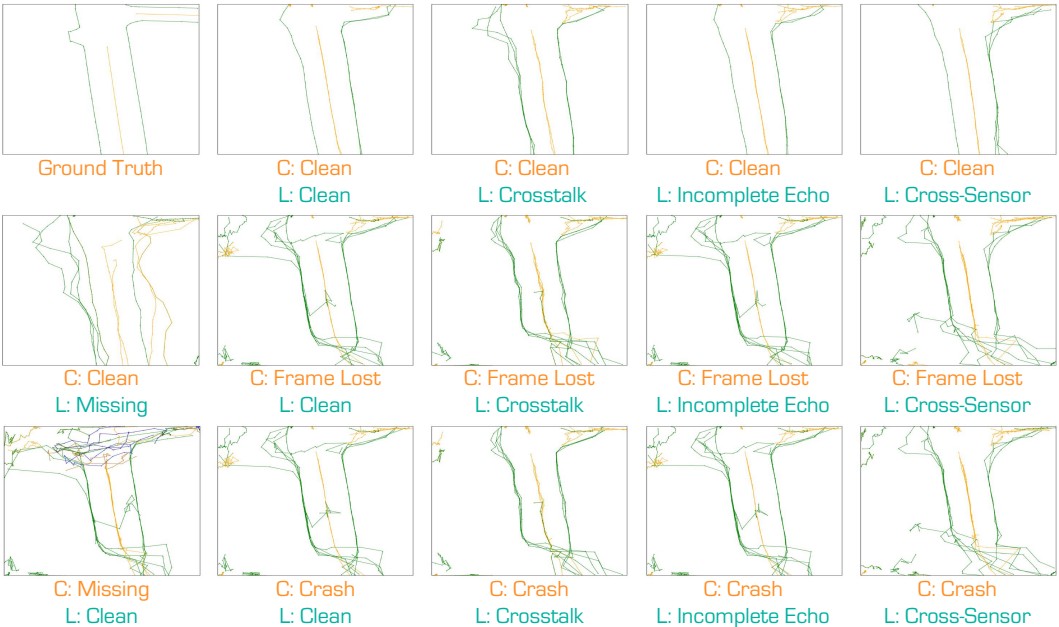

Figure 5: Qualitative assessment of camera-LiDAR fusion-based HD map construction under the Camera and LiDAR combined sensor corruptions. Kindly refer to Sec. F for additional examples.

2) Most models exhibit negligible performance drops under `Incomplete Echo`. This corruption type primarily affects data from vehicles or objects with dark colors [27], whereas the HD map construction task concerns more on static map elements. Besides, although VectorMapNet [43] achieves the best `mRR` metric, it is not less accurate in terms of `mAP` compared to HIMap [77].

## 4.4 Camera-LiDAR Fusion Benchmarking Results

To systematically evaluate the reliability of camera-LiDAR fusion-based methods, we design 13 types of multi-sensor corruptions that perturb camera and LiDAR inputs separately or concurrently. The results are presented in Fig. 4. Our findings indicate that the camera-LiDAR fusion model exhibits varying degrees of performance declines on different corruption combinations. The experimental results reveal several interesting findings, and we provide detailed analyses as follows:

1) In scenarios where Camera data is missing, the `mAP` of MapTR [41] and HIMap [77] dropped by 40.0% and 68.9%, respectively, posing a significant threat to safe perceptions. Besides, `Frame Lost` causes a worse effect than `Camera Crash` in the performance of sensor fusion-based methods. These observations verify that camera sensor failures significantly threaten HD map fusion models.

2) In scenarios where LiDAR data is missing, the `mAP` of MapTR [41] and HIMap [77] dropped by 42.1% and 41.5%, respectively, showing the indispensability of the LiDAR sensor. Moreover, the LiDAR `Crosstalk` and `Cross-Sensor` corruptions affect the performance of camera-LiDAR fusion the most. In contrast, the LiDAR `Incomplete Echo` corruption does not show a substantial impact on model performance, which is consistent with the observation under LiDAR-only configurations.

3) The results of Camera-LiDAR combined corruptions lead to worse performance than its both single-modality counterparts, highlighting the significant threats posed by both camera and LiDAR sensor failures to HD map construction tasks. Moreover, regardless of the type of LiDAR corruption combined, `Frame Lost` has a more significant impact on the fusion model performance than `Camera Crash`, underscoring the importance of multi-view inputs from the camera sensor. Among the three types of LiDAR corruptions, `Cross-Sensor` corruption affects the fusion model performance the most. This pattern remains consistent even when combined with various types of camera corruptions, illustrating the substantial threat posed by cross-configuration or cross-device LiDAR data input. We provide some qualitative examples of HD map construction under various camera-LiDAR corruption combinations in Fig. 5, which shows the performance decline under various corruptions.

Table 4: Ablation on the use of backbone nets.

| Method | Back | $AP_p.$ | $AP_d.$ | $AP_b.$ | mAP | mRR | mCE |
|---|---|---|---|---|---|---|---|
| PivotNet ○ | R50 | 53.8 | 58.8 | 59.6 | 57.4 | 45.2 | 100.0 |
| PivotNet ○ | Effi-B0 | 53.9 | 59.7 | 61.0 | 58.2 | 49.9 | 87.4 |
| PivotNet ● | SwinT | 58.7 | 63.8 | 64.9 | 62.5 | 50.8 | 77.8 |
| BeMapNet ○ | R50 | 57.7 | 62.3 | 59.4 | 59.8 | 50.3 | 100.0 |
| BeMapNet ○ | Effi-B0 | 56.0 | 62.2 | 59.0 | 59.1 | 53.9 | 94.0 |
| BeMapNet ● | SwinT | 61.3 | 64.4 | 61.6 | 62.5 | 57.9 | 75.9 |

Table 5: Ablation on different training epochs.

| Method | Epoch | $AP_p.$ | $AP_d.$ | $AP_b.$ | mAP | mRR | mCE |
|---|---|---|---|---|---|---|---|
| MapTR ○ | 24 | 46.3 | 51.5 | 53.1 | 50.3 | 49.3 | 100.0 |
| MapTR ● | 110 | 56.2 | 59.8 | 60.1 | 58.7 | 49.3 | 80.9 |
| PivotNet ○ | 30 | 58.7 | 63.8 | 64.9 | 62.5 | 50.8 | 100.0 |
| PivotNet ● | 110 | 62.6 | 68.0 | 69.7 | 66.8 | 49.9 | 90.2 |
| BeMapNet ○ | 30 | 61.3 | 64.4 | 61.6 | 62.5 | 57.9 | 100.0 |
| BeMapNet ● | 110 | 64.6 | 68.9 | 67.5 | 67.0 | 56.7 | 89.2 |

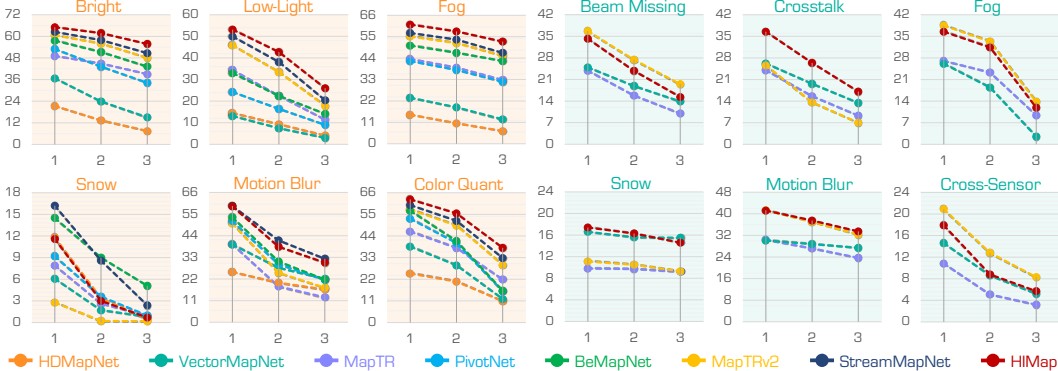

Figure 6: The mAP metrics of state-of-the-art HD map constructors under each of the three severity levels (Esay, Moderate, and Hard) in different Camera and LiDAR sensor corruption scenarios.

It is worth noting that although the performance of HIMap [77] is better than that of MapTR [41] under "clean" conditions, its robustness under corruption is relatively poorer. These observations necessitate further research focused on enhancing the robustness of camera-LiDAR fusion methods, especially when one sensory modality is absent or both the camera and LiDAR are corrupted.

## 5 Observation & Discussion

In this section, we analyze and discuss the impact of different model configurations and techniques that affect the robustness of HD map constructors, including different backbone networks, BEV encoders, temporal information, training epochs, data augmentation enhancement, and so on.

**Backbones.** We first comprehensively investigate the impact of backbone networks, with results presented in Tab. 4. Specifically, we use three different backbones in PivotNet [11] and BeMapNet [50], respectively. The results demonstrate that Swin Transformer [45] significantly retains model robustness. As an example, compared with ResNet-50 [18], the Swin Transformer [45] backbone improves the mCE of PivotNet [11] and BeMapNet [50] with 22.2% and 24.1% absolute gains, respectively. The results demonstrate that larger pretrained models tend to help enhance the robustness of feature extraction under out-of-domain data, which is in line with the observation drawn in [19, 2, 13, 10, 57].

**Different BEV Encoders.** We study several popular 2D-to-BEV transformation methods and show the results in Tab. 2. Specifically, we adopt the BEVFormer [39], BEVPool [46], and GKT [5] for the camera-only MapTR [41] model. The results demonstrate that MapTR [41] is compatible with various 2D-to-BEV methods and achieves stable robustness performance. Moreover, the mRR results of BEVPool [46] are inferior to those of BEVFormer [39] and GKT [5], verifying the effectiveness of transformer-based BEV encoders on improving HD map model robustness. GKT [5] achieves the best mCE, which is possibly due to the integration of both geometry and view transformer methods.

**Temporal Information.** We investigate the impact of utilizing temporal cues on the robustness of HD map models and show the results in Tab. 3. We examine two variants of StreamMapNet [73]: one with and one without the temporal fusion module. The results demonstrate that the temporal fusion module can significantly enhance the robustness. The mAP results here differ from those in Tab. 1 since StreamMapNet [73] was retrained following the default settings of a new train/validation split, whereas the results in Tab. 1 were obtained using the old train/validation split. It can be observed that the model with temporal cues achieves 8.4% and 14.1% absolute gains on the mRR and mCE metrics,

Table 6: Efficacy of Camera-based data augmentation techniques on HD map model robustness.

| Method | $AP_{p.}$ | $AP_{d.}$ | $AP_{b.}$ | mAP | mRR | mCE |
|---|---|---|---|---|---|---|
| None | 45.6 | 50.1 | 52.3 | 49.3 | 41.1 | 100.0 |
| Rotate [37] | 44.6 | 50.5 | 54.0 | 49.7 | 38.1 | 105.1 |
| Flip [37] | 44.7 | 53.0 | 53.4 | 50.4 | 38.7 | 102.5 |
| PhotoMetric [33] | **46.3** | **51.5** | 53.1 | 50.3 | **49.3** | **84.5** |

Table 7: Efficacy of LiDAR-based data augmentation techniques on HD map model robustness.

| Method | $AP_{p.}$ | $AP_{d.}$ | $AP_{b.}$ | mAP | mRR | mCE |
|---|---|---|---|---|---|---|
| None | 26.6 | 31.7 | 41.8 | 33.4 | 55.1 | 100.0 |
| Dropout [9] | 28.4 | 31.0 | 42.5 | 33.9 | 56.9 | 98.9 |
| RTS-LiDAR [23] | 28.3 | 32.7 | 44.1 | 35.0 | 57.0 | 94.0 |
| PolarMix [61] | **30.1** | **33.0** | **46.1** | **36.4** | 55.2 | 93.5 |

respectively. This verifies that temporal fusion can provide additional complementary information under sensor corruptions, thereby enhancing robustness against different sensor corruptions.

**Training Epochs.** In this setting, we study three HD map models (MapTR [41], PivotNet [11], and BeMapNet [50]) trained with different numbers of epochs, with results shown in Tab. 5. It can be observed that training for more epochs significantly improves both performance on the "clean" set and robustness to corruptions. For example, utilizing a longer training schedule enhances robustness in mCE metrics: MapTR [41] (+19.1%), PivotNet [11] (+9.8%), and BeMapNet [50] (+10.8%). Notably, the performance of these models on the "clean" set also improves as the training schedule lengthens, suggesting that extended training allows the model to better learn the inherent patterns from the dataset, thereby achieving better generalization performance on corrupted data [21].

**Data Augmentations to Boost Corruption Robustness.** We investigate the effect of various data augmentation techniques on the robustness of HD map models. As multi-modal data augmentation remains an open issue, in this work, we focus on investigating the effects of image and LiDAR data augmentation techniques. Specifically, we study three distinct image data augmentation methods, *i.e.* Rotate [37], Flip [37], and PhotoMetric [33], and three distinct LiDAR-based data augmentation methods, *i.e.* Dropout [9], RTS-LiDAR (Rotate-Translate-Scale for LiDAR) [23], and PolarMix [61].

1) For Camera-based data augmentations, we choose MapTR-R50 [41] as the baseline and show results in Tab. 6. It can be observed that image augmentation methods moderately improve model performance on the "clean" set. However, they do not consistently enhance model robustness. For example, PhotoMetric [33] improves the robustness metrics, mRR and mCE, by 8.2% and 15.5%, respectively, whereas Rotate [37] and Flip [37] weaken the robustness. This discrepancy likely arises from the fact that PhotoMetric [33] functions similarly to corruption augmentation for certain types, such as Bright and Low-Light, differing from other augmentation methods.

2) For LiDAR-based data augmentations, we choose the MapTR-LiDAR [41] model due to its superior robustness among all LiDAR-only models. The results of different LiDAR augmentations are presented in Tab. 7. We observe that all LiDAR augmentations significantly improve the model performance on the "clean" set. In particular, PolarMix [61] achieves a 3.0% absolute performance gain. Moreover, all LiDAR augmentation techniques are effective in enhancing the model robustness, reducing the absolute mCE values by 1.1% for Dropout [9], 6.0% for RTS-LiDAR [23], and 6.5% for PolarMix [61], respectively. These results demonstrate the effectiveness of LiDAR augmentation methods in improving the corruption robustness of LiDAR-only HD map construction methods.

# 6 Conclusion

In this work, we conducted the first study of benchmarking and analyzing the reliability of HD map construction methods under sensor corruptions that occur in real-world driving environments. Our results reveal key factors that coped closely with the out-of-domain robustness, highlighting crucial aspects in retaining satisfactory accuracy. We hope our comprehensive benchmarks, in-depth analyses, and insightful findings could help better understand the robustness of HD map construction tasks and offer useful insights into designing more reliable HD map constructors in future studies.

**Potential Limitation.** While our benchmark encompasses an abundant number of sensor corruption types, it is hard to cover the entirety of out-of-distribution contexts in real-world applications due to their unpredictable complexity. Furthermore, our experiments confirm the efficacy of standard data augmentation techniques in enhancing robustness, offering promising results. Nonetheless, further explorations into more advanced methods and network designs are warranted for future research.

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

# Appendix

This technical appendix provides additional details of the proposed MapBench, as well as experimental results that are omitted from the main body of this paper due to the page limit.

Specifically, this appendix is organized as follows:

- Sec. A presents the detailed definitions of our sensor corruption types.
- Sec. B provides additional implementation details of multi-sensor corruptions.
- Sec. C presents additional results on the temporally-aggregate LiDAR-only benchmark.
- Sec. D offers detailed experimental results in terms of the class-wise CE and RR scores for camera-based and LiDAR-based HD map construction models.
- Sec. E provides the full benchmark configurations.
- Sec. F displays additional qualitative examples of HD map construction under the camera and LiDAR sensor corruptions.
- Sec. G discusses the limitation and potential societal impact of this work.
- Sec. H follows the NeurIPS Dataset & Benchmark guideline to document necessary information about the proposed datasets and benchmarks.
- Sec. I acknowledges the use of public resources, during the course of this work.

## A  Sensor Corruption Definition

In this section, we provide detailed descriptions and configurations of the camera and LiDAR sensor corruptions used in our benchmark. These corruptions are designed to simulate various real-world conditions that autonomous driving systems may encounter.

### A.1  Camera Sensor Corruptions

We detail the descriptions and severity level setups for 8 types of camera sensor corruptions [62] in Tab. 8. These corruptions are:

- Bright and Low-Light: Simulate various lighting conditions to test the robustness of HD map constructors in different illumination scenarios.
- Fog and Snow: Represent visually obstructive forms of precipitation, simulating extreme weather conditions that can obscure the camera's view.
- Color Quantization: Reduces the number of colors in an image while preserving its overall visual appearance, challenging the model's ability to handle color variations.
- Motion Blur: Occurs when the camera moves quickly, causing blurring in the captured images.
- Camera Crash: Simulates continuous loss of images from certain viewpoints due to camera failure.
- Frame Lost: Represents random loss of frames over time, testing the model's resilience to intermittent data loss.

Visualization examples of camera sensor corruptions under different severity levels (Easy, Moderate, and Hard) are shown in Figure 7.

### A.2  LiDAR Sensor Corruptions

The detailed descriptions and severity level setups for 8 types of LiDAR corruptions [27] are illustrated in Table 9. These corruptions include:

- Fog: Causes back-scattering and attenuation of LiDAR points, simulating foggy weather conditions.

Table 8: Definitions and severity level setups for the Camera sensor corruption simulations in the proposed MapBench. A total of 8 distinct types of camera corruption are illustrated, including [1]Bright, [2]Low-Light (Dark), [3]Fog, [4]Snow, [5]Color Quantization (Quant), [6]Motion Blur (Motion), [7]Camera Crash (Camera), and [8]Frame Lost (Frame).

| Type | Description | Parameter | Easy | Moderate | Hard |
|------|-------------|-----------|------|----------|------|
| Bright | varying daylight intensity | adjustment in HSV space | 0.2 | 0.4 | 0.5 |
| Dark | varying daylight intensity | scale factor | 0.5 | 0.4 | 0.3 |
| Fog | a visually obstructive form of precipitation | (thickness, smoothness) | (2.0, 2.0) | (2.5, 1.5) | (3.0, 1.4) |
| Snow | a visually obstructive form of precipitation | (mean, std, scale, threshold, blur radius, blur std, blending ratio) | (0.1, 0.3, 3.0, 0.5, 10.0, 4.0, 0.8) | (0.2, 0.3, 2, 0.5, 12, 4, 0.7) | (0.55, 0.3, 4, 0.9, 12, 8, 0.7) |
| Quant | reducing the number of colors | bit number | 5 | 4 | 3 |
| Motion | moving camera quickly | (radius, sigma) | (15, 5) | (15, 12) | (20, 15) |
| Camera | dropping view images | number of dropped cameras | 2 | 4 | 5 |
| Frame | dropping temporal frames | probability of frame dropping | 2/6 | 4/6 | 5/6 |

- Wet Ground: Results in significantly attenuated laser echoes due to water height and mirror refraction rate.
- Snow: Similar to Fog, it leads to back-scattering and attenuation of LiDAR points.
- Motion Blur: Caused by vehicle movement, blurring the LiDAR point cloud.
- Beam Missing: Simulates the loss of certain laser beams due to occlusion by dust and insects.
- Crosstalk: Creates noisy points within the mid-range areas between two (or multiple) sensors, simulating interference.
- Incomplete Echo: Represents incomplete LiDAR readings in some scan echoes.
- Cross-Sensor: Arises due to the large variety of LiDAR sensor configurations (e.g., beam number, field-of-view, and sampling frequency).

Visualization examples of LiDAR sensor corruptions under different severity levels (Easy, Moderate, and Hard) are shown in Figure 8.

## B  Multi-Sensor Corruptions

In this section, we provide detailed descriptions and configurations of the combined camera-LiDAR sensor corruptions used in our benchmark. These combined corruptions simulate scenarios where both camera and LiDAR sensors are simultaneously affected by adverse conditions, providing a comprehensive evaluation of the robustness of camera-LiDAR fusion models.

### B.1  Camera-Only Corruptions

For Camera-only corruptions, we design three combinations to evaluate the impact on models when only the camera input is affected:

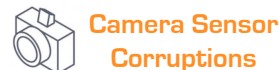

**Camera Sensor Corruptions**

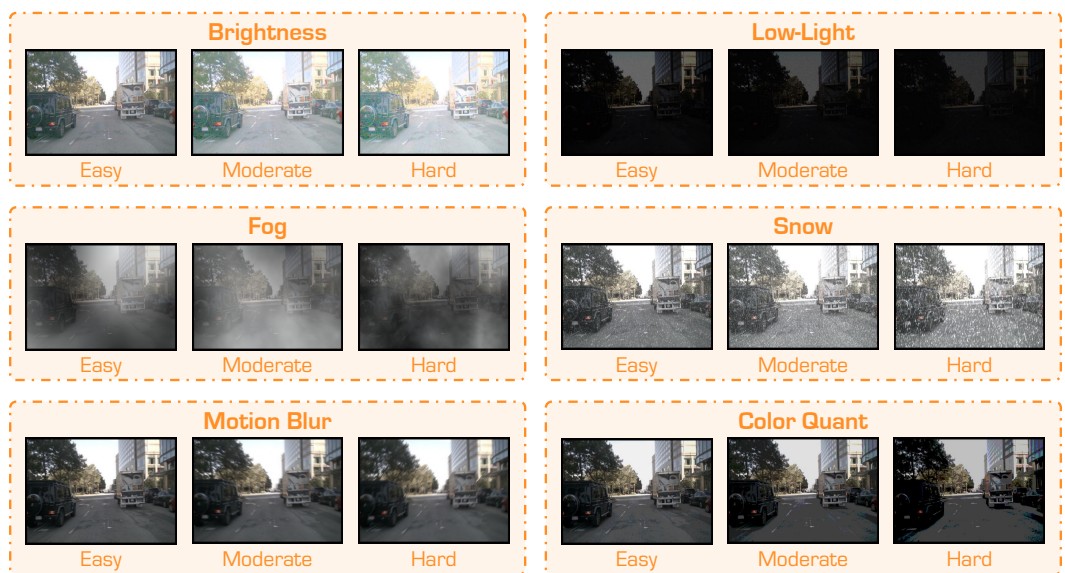

Figure 7: Visualizations of different Camera sensor corruptions under three severity levels, *i.e.*, Easy, Moderate, and Hard, in our benchmark. Best viewed in color and zoomed in for details.

1. `Unavailable Camera and Clean LiDAR Data`: This scenario simulates a complete failure of the camera sensor while the LiDAR sensor remains fully operational.

2. `Camera Crash and Clean LiDAR Data`: In this setup, the camera experiences intermittent crashes, leading to continuous loss of images from certain viewpoints, while LiDAR data remains unaffected.

3. `Camera Frame Lost and Clean LiDAR Data`: This corruption simulates random loss of camera frames over time, with the LiDAR sensor providing clean data.

The results of these experiments are shown in Tab. 10 (a) and Tab. 11 (a). Our findings indicate that camera-LiDAR fusion models exhibit varying degrees of performance decline under different camera-only corruption scenarios.

Specifically, when Camera data is completely unavailable, the `mAP` of MapTR [41] and HIMap [77] dropped by 40.0% and 68.9%, respectively, highlighting the significant impact of camera sensor failure on safe perception. Moreover, `Frame Lost` causes a more severe performance degradation compared to `Camera Crash` in fusion-based methods. For instance, when `Frame Lost` corruption occurs, the absolute decreases in the `mAP` metrics of MapTR [41] and HIMap [77] are 26.2% and 48.1%, respectively. These observations underscore the vulnerability of HD map fusion models to camera sensor failures.

## B.2 LiDAR-Only Corruptions

For LiDAR-only corruptions, we design four combinations to assess the impact when only the LiDAR input is affected:

1. `Unavailable LiDAR and Clean Camera Data`: This scenario simulates a complete failure of the LiDAR sensor while the camera sensor remains fully operational.

2. `LiDAR Incomplete Echo and Clean Camera Data`: This setup simulates incomplete LiDAR readings in some scan echoes, with the camera providing clean data.

3. `LiDAR Crosstalk and Clean Camera Data`: In this configuration, the LiDAR sensor experiences crosstalk, creating noisy points within the mid-range areas between multiple sensors, while the camera data remains clean.

Table 9: Definitions and severity level setups for the LiDAR sensor corruption simulations in the proposed MapBench. A total of 8 distinct types of LiDAR corruption are illustrated, including [1]`Fog`, [2]`Wet Ground` (Wet), [3]`Snow`, [4]`Motion Blur` (Motion), [5]`Beam Missing` (Beam), [6]`Crosstalk`, [7]`Incomplete Echo` (Echo), and [8]`Cross-Sensor` (Sensor).

| Type | Description | Parameter | Easy | Moderate | Hard |
|---|---|---|---|---|---|
| Fog | back-scattering and attenuation of LiDAR points | beta | 0.008 | 0.05 | 0.2 |
| Wet | significantly attenuated laser echoes | (water height, noise floor) | (0.2, 0.2) | (1.0, 0.3) | (1.2, 0.7) |
| Snow | back-scattering and attenuation of LiDAR points | (snowfall rate, terminal velocity) | (0.5, 2.0) | (1.0, 1.6) | (2.5, 1.6) |
| Motion | blur caused by vehicle movement | trans std | 0.2 | 0.3 | 0.4 |
| Beam | loss of certain light impulses | beam number to drop | 8 | 16 | 24 |
| Crosstalk | light impulses interference | percentage | 0.03 | 0.07 | 0.12 |
| Echo | incomplete LiDAR readings | drop ratio | 0.75 | 0.85 | 0.95 |
| Sensor | cross sensor data | beam number to drop | 8 | 16 | 20 |

4. `LiDAR Cross-Sensor and Clean Camera Data`: This corruption simulates cross-sensor issues due to varying LiDAR sensor configurations (e.g., beam number, field-of-view, and sampling frequency), with the camera data being clean.

The results are presented in Tab. 10 (b) and Tab. 11 (b). When LiDAR data is completely unavailable, the `mAP` of MapTR [41] and HIMap [77] dropped by 42.1% and 41.5%, respectively, demonstrating the critical importance of LiDAR sensors in HD map construction. Additionally, `LiDAR Cross-Sensor` and `Crosstalk` corruptions have the most significant impact on the performance of camera-LiDAR fusion models. For instance, the `mAP` metrics for `Cross-Sensor` and `Crosstalk` show absolute decreases of 22.9% and 21.0% in the MapTR model, respectively. In contrast, the `LiDAR Incomplete Echo` corruption does not substantially impact model performance, aligning with observations under LiDAR-only configurations.

### B.3 Camera-LiDAR Corruption Combinations

We design six types of combined Camera-LiDAR corruption scenarios that perturb both sensor inputs concurrently, using the previously mentioned image and LiDAR sensor failure types:

1. `Unavailable Camera and Unavailable LiDAR`: Both camera and LiDAR sensors are completely unavailable.

2. `Camera Crash and LiDAR Crosstalk`: Simulates intermittent camera crashes and LiDAR crosstalk.

3. `Camera Frame Lost and LiDAR Incomplete Echo`: Represents random loss of camera frames and incomplete LiDAR echoes.

4. `Low-Light Camera and LiDAR Cross-Sensor`: Combines low-light conditions for the camera and cross-sensor issues for LiDAR.

5. `Motion Blur Camera and LiDAR Motion Blur`: Both camera and LiDAR sensors experience motion blur.

6. `Foggy Camera and Foggy LiDAR`: Both sensors are affected by fog.

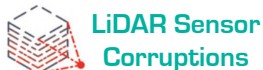

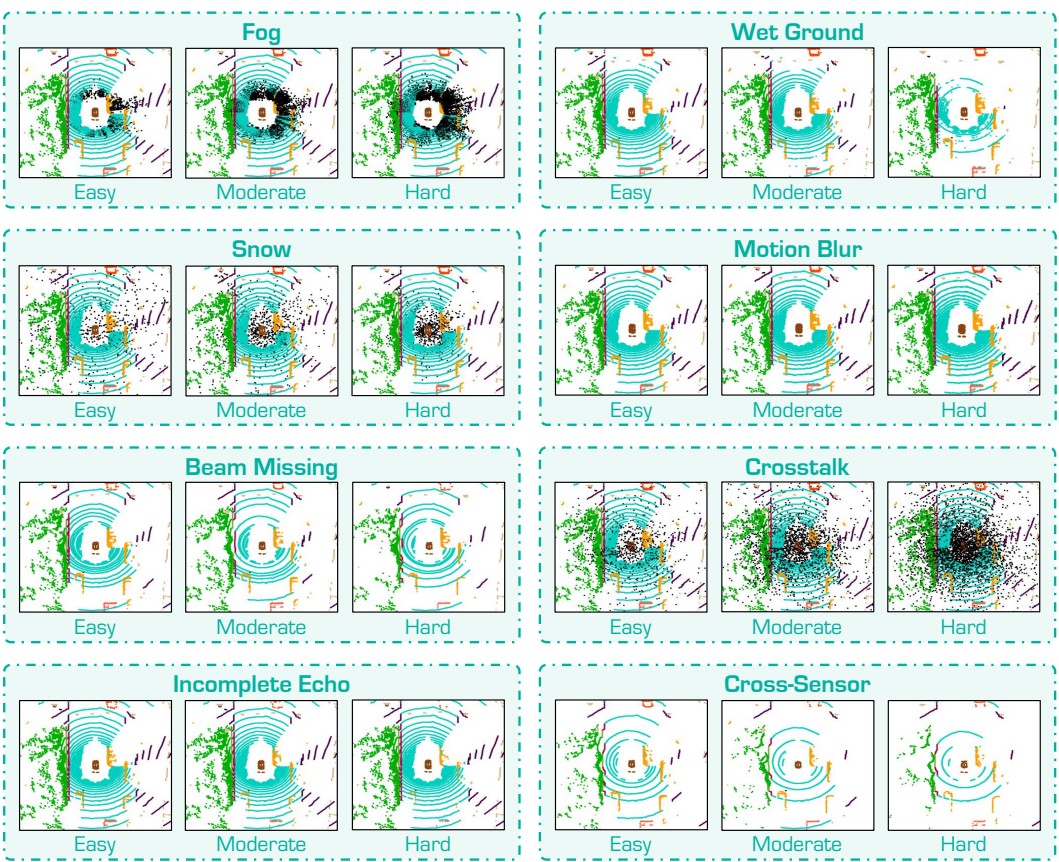

Figure 8: Visualizations of different LiDAR sensor corruptions under three severity levels, *i.e.*, `Easy`, `Moderate`, and `Hard`, in our benchmark. Best viewed in color and zoomed in for details.

The experimental results are shown in Tab. 10 (c) and Tab. 11 (c). The results indicate that combined camera-LiDAR corruptions generally result in more severe performance degradation compared to camera-only or LiDAR-only corruptions, demonstrating the compounded threats of sensor failures to HD map construction tasks.

Moreover, in scenarios involving camera corruptions, `Frame Lost` has a significantly worse impact on fusion model performance than `Camera Crash`, highlighting the importance of continuous multi-view inputs from the camera sensor. This impact is consistent across various LiDAR corruption types. Similarly, among the LiDAR corruptions, `Cross-Sensor` affects fusion model performance the most, irrespective of the camera corruption type, underscoring the substantial threat posed by cross-configuration or cross-device LiDAR data input.

As shown in Tab. 10 (a)-(c) and Tab. 11 (a)-(c), camera-LiDAR fusion models consistently exhibit superior robustness to corruptions compared to single-modality models, regardless of whether one or both modalities are corrupted. These findings highlight the necessity for further research focused on enhancing the robustness of HD map camera-LiDAR fusion models, especially when one sensory modality is compromised or both are affected by adverse conditions.

## C  Additional Results of Temporally-Aggregated LiDAR-Only Benchmark

In this section, we report the robustness of LiDAR sensor corruptions using temporally aggregated LiDAR points as the input for three LiDAR-only HD map models. The detailed results are presented

Table 10: The results of the MapTR [41] model under different model configurations and multi-sensor corruptions in MapBench.

| Method | Modality | Camera | LiDAR | $\text{AP}_{ped.}$ | $\text{AP}_{div.}$ | $\text{AP}_{bou.}$ | mAP |
|---|---|---|---|---|---|---|---|
| MapTR [41] | C & L | ✓ | ✓ | 55.9 | 62.3 | 69.3 | 62.5 |
| MapTR [41] | C | ✓ | − | 46.3 | 51.5 | 53.1 | 50.3 |
| MapTR [41] | C | Camera Crash | − | 18.0 | 14.5 | 12.4 | 15.0 |
| MapTR [41] | C | Frame Lost | − | 13.9 | 15.1 | 13.4 | 14.2 |
| (a) MapTR [41] | C & L | ✗ | ✓ | 15.0 | 18.2 | 34.4 | 22.5 |
| MapTR [41] | C & L | Camera Crash | ✓ | 32.5 | 36.5 | 48.4 | 39.1 |
| MapTR [41] | C & L | Frame Lost | ✓ | 29.1 | 33.7 | 46.1 | 36.3 |
| MapTR [41] | L | − | ✓ | 26.6 | 31.7 | 41.8 | 33.4 |
| MapTR [41] | L | − | Incomplete Echo | 26.3 | 29.9 | 40.6 | 32.3 |
| MapTR [41] | L | − | Crosstalk | 13.6 | 15.0 | 20.3 | 16.3 |
| MapTR [41] | L | − | Cross-Sensor | 3.5 | 6.6 | 8.9 | 6.4 |
| (b) MapTR [41] | C & L | ✓ | ✗ | 20.7 | 27.4 | 13.1 | 20.4 |
| MapTR [41] | C & L | ✓ | Incomplete Echo | 47.9 | 55.2 | 62.2 | 55.1 |
| MapTR [41] | C & L | ✓ | Crosstalk | 36.7 | 42.5 | 45.3 | 41.5 |
| MapTR [41] | C & L | ✓ | Cross-Sensor | 33.9 | 42.9 | 42.0 | 39.6 |
| MapTR [41] | C & L | Camera Crash | Incomplete Echo | 32.4 | 35.6 | 47.8 | 38.6 |
| MapTR [41] | C & L | Camera Crash | Crosstalk | 19.7 | 21.6 | 26.9 | 22.7 |
| (c) MapTR [41] | C & L | Camera Crash | Cross-Sensor | 18.4 | 20.8 | 23.2 | 20.8 |
| MapTR [41] | C & L | Frame Lost | Incomplete Echo | 28.9 | 32.8 | 45.5 | 35.8 |
| MapTR [41] | C & L | Frame Lost | Crosstalk | 16.9 | 19.9 | 25.5 | 20.8 |
| MapTR [41] | C & L | Frame Lost | Cross-Sensor | 15.8 | 19.4 | 22.2 | 19.1 |

in Tab. 12 and Tab. 13. Notably, each table lists two mean Average Precision (mAP) values for each model: the first value corresponds to our re-trained model, and the second value is directly sourced from the original paper. Our re-trained models generally perform better than or on par with the originally reported results, validating the effectiveness of our re-training process. Since the authors of the original models have not shared their pre-trained models, our re-trained versions are utilized in all subsequent experiments.

To simulate corruptions, we independently corrupt each LiDAR frame and then temporally aggregate the corrupted frames, mirroring the aggregation process used for clean data. However, it is important to note that this method does not ensure temporal consistency, introducing a potential bias compared to real-world corruptions. Temporally-aggregated LiDAR data were generated for five types of corruptions: Fog, Motion Blur, Beam Missing, Crosstalk, and Cross-Sensor. The remaining three corruption types were not generated due to the unavailability of necessary information, such as semantic labels for the LiDAR points.

Tab. 12 and Tab. 13 reveal that LiDAR Crosstalk and Cross-Sensor corruptions have the most significant impact on the performance of LiDAR-only models. Consistent with observations from single-frame LiDAR points, Cross-Sensor corruption impairs the models the most, highlighting the substantial threat posed to the robustness of LiDAR-only HD map models. The LiDAR Cross-Sensor corruption demonstrates that the domain gap caused by variations in LiDAR configurations and devices significantly reduces model performance.

Moreover, the use of temporally inconsistent aggregated data does not fully align with real-world scenarios, indicating an open issue in the generation of multi-moment LiDAR corruption data. Addressing this gap is crucial for developing more realistic and effective benchmarks for evaluating the robustness of LiDAR-only HD map models under temporal aggregation.

# D   Class-Wise CE and RR Results for Camera and LiDAR Models

In this section, we present detailed experimental results in terms of class-wise Calibration Error (CE) and Robustness Ratio (RR) for camera-based and LiDAR-based HD map construction models, as shown in Tab. 14 to Tab. 17. Based on the empirical evaluation results, we derive several important findings, summarized as follows:

Table 11: The results of the HIMap [77] model under different model configurations and multi-sensor corruptions in MapBench.

| | Method | Modality | Camera | LiDAR | $AP_{ped.}$ | $AP_{div.}$ | $AP_{bou.}$ | mAP |
|---|---|---|---|---|---|---|---|---|
| | HIMap [77] | C & L | ✓ | ✓ | 71.0 | 72.4 | 79.4 | 74.3 |
| | HIMap [77] | C | ✓ | – | 62.2 | 66.5 | 67.9 | 65.5 |
| | HIMap [77] | C | Camera Crash | – | 27.3 | 19.4 | 11.6 | 19.4 |
| | HIMap [77] | C | Frame Lost | – | 21.7 | 19.1 | 16.1 | 19.0 |
| (a) | HIMap [77] | C & L | ✗ | ✓ | 40.9 | 46.4 | 74.7 | 50.7 |
| | HIMap [77] | C & L | Camera Crash | ✓ | 36.3 | 27.7 | 20.9 | 28.3 |
| | HIMap [77] | C & L | Frame Lost | ✓ | 29.9 | 25.0 | 23.8 | 26.2 |
| | HIMap [77] | L | – | ✓ | 54.8 | 64.7 | 73.5 | 64.3 |
| | HIMap [77] | L | – | Incomplete Echo | 35.4 | 41.1 | 52.7 | 43.1 |
| | HIMap [77] | L | – | Crosstalk | 20.9 | 23.8 | 35.3 | 26.7 |
| | HIMap [77] | L | – | Cross-Sensor | 7.8 | 10.2 | 14.4 | 10.8 |
| (b) | HIMap [77] | C & L | ✓ | ✗ | 30.7 | 38.7 | 29.0 | 32.8 |
| | HIMap [77] | C & L | ✓ | Incomplete Echo | 59.1 | 63.7 | 69.9 | 64.2 |
| | HIMap [77] | C & L | ✓ | Crosstalk | 54.1 | 57.5 | 63.4 | 58.3 |
| | HIMap [77] | C & L | ✓ | Cross-Sensor | 44.2 | 50.7 | 50.8 | 48.5 |
| | HIMap [77] | C & L | Camera Crash | Incomplete Echo | 36.2 | 26.9 | 20.5 | 27.9 |
| | HIMap [77] | C & L | Camera Crash | Crosstalk | 29.2 | 19.3 | 12.9 | 20.5 |
| (c) | HIMap [77] | C & L | Camera Crash | Cross-Sensor | 23.1 | 13.8 | 5.9 | 14.3 |
| | HIMap [77] | C & L | Frame Lost | Incomplete Echo | 29.9 | 24.4 | 23.5 | 25.9 |
| | HIMap [77] | C & L | Frame Lost | Crosstalk | 23.6 | 18.9 | 18.0 | 20.2 |
| | HIMap [77] | C & L | Frame Lost | Cross-Sensor | 17.7 | 14.3 | 11.2 | 14.4 |

1) For camera sensor corruptions, `Snow` corruption significantly degrades model performance, posing a major threat to autonomous driving as snow covers the road, rendering map elements unrecognizable. Additionally, `Frame Lost` and `Camera Crash` corruptions are highly challenging for all models, underscoring the serious threats posed by camera sensor failures to camera-only HD map models.

2) For LiDAR sensor corruptions, `Snow` and `Cross-Sensor` corruptions notably impact robustness performance. This indicates that weather conditions and sensor failure corruptions pose significant threats to the robustness of LiDAR-based HD map models. However, most models exhibit negligible performance drops under `Incomplete Echo` corruption, primarily due to the minimal relevance between this type of corruption and the HD map construction task.

Overall, our findings highlight that `Snow` corruption among all camera and LiDAR corruptions significantly degrades model performance. This corruption obscures the road, rendering map elements unrecognizable and posing a substantial threat to autonomous driving. Additionally, sensor failure corruptions, such as `Frame Lost` and `Incomplete Echo`, present serious challenges for all models, demonstrating the critical threats that sensor failures pose to HD map models.

# E    Full Benchmark Configurations

In this section, we provide the complete benchmarking results of the studied models. We report the basic information for each model in Tab. 18, including input modality, BEV encoder, backbone, training epochs, and their performance on the clean nuScenes validation set. The detailed benchmarking results are shown in Tab. 19 to Tab. 23.

Generally, models with higher accuracy on the clean set tend to achieve better corruption robustness. Specifically, StreamMapNet [73] and HIMap [77] demonstrate the best robustness among camera-only and LiDAR-only models, respectively. However, the overall robustness across all models remains relatively low.

We hope that our comprehensive benchmarks, in-depth analyses, and insightful findings will help researchers better understand the robustness challenges in HD map construction tasks and provide valuable insights for designing more reliable HD map constructors in future studies.

Table 12: The CE (Corruption Error) of **LiDAR-only** HD map models (**taking temporally-aggregated LiDAR points as input**) in MapBench. Underlined values are directly from the original paper.

| Method | mAP | mCE | Fog | Motion | Beam | Crosstalk | Sensor |
|---|---|---|---|---|---|---|---|
| MapTR [41] | 56.2 / 55.6 | 100.0 | 100.0 | 100.0 | 100.0 | 100.0 | 100.00 |
| VectorMapNet [43] | 40.5 / 34.0 | 124.4 | 179.4 | 63.3 | 190.5 | 71.1 | 117.5 |
| MapTRv2 [42] | 61.0 / 61.5 | 89.7 | 80.9 | 114.5 | 61.3 | 116.8 | 74.8 |
| HIMap [77] | 64.3 / 64.3 | 70.1 | 63.3 | 77.0 | 54.3 | 83.1 | 73.1 |

Table 13: The RR (Resilience Rate) of **LiDAR-only** HD map models (**taking temporally-aggregated LiDAR points as input**) in MapBench. Underlined values are directly from the original paper.

| Method | mAP | mRR | Fog | Motion | Beam | Crosstalk | Sensor |
|---|---|---|---|---|---|---|---|
| MapTR [41] | 56.2 / 55.6 | 49.7 | 66.5 | 33.5 | 90.3 | 18.0 | 40.0 |
| VectorMapNet [43] | 40.5 / 34.0 | 65.5 | 61.0 | 75.7 | 96.3 | 49.1 | 45.3 |
| MapTRv2 [42] | 61.0 / 61.5 | 48.9 | 66.9 | 25.0 | 95.4 | 9.2 | 48.3 |
| HIMap [77] | 64.3 / 64.3 | 54.5 | 70.2 | 39.3 | 93.1 | 23.4 | 46.4 |

# F    Qualitative Assessments

In this section, we provide additional qualitative examples of HD map construction under various camera and LiDAR sensor corruptions in Fig. 9 - Fig. 17. These examples offer a visual comparison of the performance of different models and highlight the impact of sensor corruptions on HD map construction tasks. We include visualizations for several corruption types, demonstrating how each type affects the perception and mapping capabilities of the models. The qualitative examples are presented for both Camera-only and LiDAR-only configurations, as well as for Camera-LiDAR fusion models. This comprehensive visual analysis aims to complement the quantitative results discussed in the main paper and provide deeper insights into the robustness of HD map construction models. Based on the qualitative results, we draw several important findings, which can be summarized as follows:

1) Among all qualitative examples of Camera and LiDAR sensor, Snow corruption significantly degrades model performance; it covers the road, rendering map elements unrecognizable and posing a major threat to autonomous driving. Besides, sensor failure corruptions (*e.g.*. Frame Lost and Incomplete Echo) are also challenging for all models, demonstrating the serious threats of sensor failures on HD map models.

2) The qualitative results of Camera-LiDAR combined corruptions lead to worse performance than its both single-modality counterparts, highlighting the significant threats posed by both camera and LiDAR sensor failures to HD map construction tasks. These observations necessitate further research focused on enhancing the robustness of camera-LiDAR fusion methods, especially when one sensory modality is absent or both the camera and LiDAR are corrupted.

# G    Limitation and Potential Societal Impact

In this section, we elaborate on the limitations and potential societal impact of this work.

## G.1    Potential Limitations

While MapBench provides a comprehensive benchmark for evaluating the robustness of HD map construction methods, there are several limitations to consider:

- **Scope of Corruptions:** Although our benchmark includes 29 types of sensor corruptions, it may not cover all possible real-world scenarios. There could be additional adverse conditions or sensor anomalies that were not included in this study, potentially limiting the generalizability of our findings.

Table 14: The CE (Corruption Error) of **camera-only** HD map models in MapBench.

| # | Method | mAP | mCE | Camera | Frame | Quant | Motion | Bright | Dark | Fog | Snow |
|---|--------|-----|-----|--------|-------|-------|--------|--------|------|-----|------|
| - | MapTR [41] | 50.3 | 100.0 | 100.0 | 100.0 | 100.0 | 100.00 | 100.0 | 100.0 | 100.0 | 100.0 |
| 1 | HDMapNet [35] | 23.0 | 187.8 | 142.1 | 137.8 | 203.2 | 114.9 | 335.5 | 165.2 | 308.0 | 95.5 |
| 2 | VectorMapNet [43] | 40.9 | 148.5 | 103.8 | 107.5 | 146.9 | 79.1 | 239.4 | 173.2 | 234.7 | 103.1 |
| 3 | PivotNet [11] | 57.4 | 96.3 | 93.0 | 90.9 | 90.8 | 62.3 | 102.7 | 127.9 | 105.3 | 97.6 |
| 4 | PivotNet [11] | 58.2 | 84.1 | 94.5 | 92.6 | 75.2 | 51.5 | 82.6 | 114.9 | 76.5 | 95.4 |
| 5 | PivotNet [11] | 62.5 | 74.7 | 90.5 | 86.1 | 67.6 | 44.4 | 64.3 | 100.5 | 55.5 | 88.9 |
| 6 | PivotNet [11] | 66.8 | 68.9 | 82.9 | 79.3 | 62.3 | 35.0 | 52.0 | 103.6 | 44.8 | 90.9 |
| 7 | BeMapNet [50] | 59.8 | 78.5 | 87.2 | 84.2 | 81.3 | 58.5 | 68.5 | 99.1 | 65.7 | 83.1 |
| 8 | BeMapNet [50] | 59.1 | 73.6 | 88.4 | 87.4 | 76.9 | 43.0 | 68.7 | 77.8 | 74.8 | 71.9 |
| 9 | BeMapNet [50] | 62.5 | 60.5 | 73.5 | 76.8 | 50.4 | 32.3 | 55.2 | 83.8 | 47.3 | 64.7 |
| 10 | BeMapNet [50] | 67.7 | 54.8 | 69.2 | 72.1 | 47.4 | 26.2 | 40.9 | 77.3 | 38.4 | 66.5 |
| 11 | MapTR [41] | 49.3 | 125.4 | 102.1 | 100.6 | 139.8 | 89.6 | 187.9 | 134.5 | 149.4 | 99.3 |
| 13 | MapTR [41] | 49.7 | 133.4 | 107.6 | 104.4 | 141.8 | 81.8 | 214.3 | 151.9 | 165.6 | 100.0 |
| 14 | MapTR [41] | 50.4 | 128.4 | 113.5 | 107.7 | 137.3 | 83.3 | 194.4 | 142.2 | 146.0 | 102.8 |
| 15 | MapTR [41] | 58.7 | 80.9 | 82.0 | 83.1 | 59.3 | 83.3 | 56.6 | 111.7 | 56.0 | 105.3 |
| 16 | MapTR [41] | 48.7 | 103.0 | 108.7 | 106.2 | 103.7 | 99.0 | 107.2 | 97.8 | 100.2 | 101.2 |
| 17 | MapTR [41] | 50.1 | 102.2 | 102.6 | 100.6 | 103.2 | 108.8 | 100.2 | 96.6 | 99.5 | 105.8 |
| 18 | MapTRv2 [42] | 61.5 | 72.6 | 87.0 | 85.1 | 58.5 | 71.1 | 52.6 | 65.9 | 51.4 | 109.0 |
| 19 | StreamMapNet [73] | 63.4 | 64.8 | 105.8 | 94.4 | 50.2 | 37.5 | 45.5 | 54.8 | 46.2 | 84.2 |
| 20 | StreamMapNet [73] | 23.8 | 183.8 | 140.2 | 134.9 | 218.5 | 156.8 | 315.1 | 155.3 | 247.7 | 102.1 |
| 21 | StreamMapNet [73] | 28.0 | 155.5 | 117.7 | 108.4 | 179.4 | 128.7 | 262.6 | 139.0 | 202.5 | 105.9 |
| 22 | HIMap [77] | 65.5 | 56.9 | 84.4 | 82.0 | 39.6 | 40.9 | 34.5 | 44.1 | 33.7 | 95.9 |

Table 15: The RR (Resilience Rate) of **camera-only** HD map models in MapBench.

| # | Method | mAP | mRR | Camera | Frame | Quant | Motion | Bright | Dark | Fog | Snow |
|---|--------|-----|-----|--------|-------|-------|--------|--------|------|-----|------|
| - | MapTR [41] | 50.3 | 49.3 | 29.9 | 28.3 | 70.7 | 47.0 | 88.7 | 45.5 | 76.9 | 7.7 |
| 1 | HDMapNet [35] | 23.0 | 43.3 | 17.4 | 19.4 | 71.7 | 79.0 | 63.6 | 35.2 | 40.4 | 19.9 |
| 2 | VectorMapNet [43] | 40.9 | 40.6 | 33.1 | 29.3 | 63.1 | 70.6 | 59.9 | 18.6 | 43.5 | 6.8 |
| 3 | PivotNet [11] | 57.4 | 45.2 | 29.9 | 29.2 | 63.7 | 59.8 | 76.1 | 29.0 | 65.5 | 8.1 |
| 4 | PivotNet [11] | 58.2 | 49.9 | 28.8 | 28.0 | 70.5 | 66.4 | 82.1 | 33.0 | 75.7 | 14.9 |
| 5 | PivotNet [11] | 62.5 | 50.8 | 28.9 | 29.1 | 69.3 | 66.5 | 83.7 | 36.6 | 80.2 | 12.1 |
| 6 | PivotNet [11] | 66.8 | 49.9 | 30.4 | 30.1 | 65.3 | 68.1 | 82.5 | 32.8 | 79.6 | 10.2 |
| 7 | BeMapNet [50] | 59.8 | 50.3 | 31.3 | 30.9 | 63.8 | 59.0 | 84.8 | 38.7 | 77.8 | 15.9 |
| 8 | BeMapNet [50] | 59.1 | 53.9 | 30.7 | 29.7 | 67.2 | 71.1 | 85.7 | 48.5 | 74.7 | 23.1 |
| 9 | BeMapNet [50] | 62.5 | 57.9 | 36.8 | 33.2 | 76.9 | 75.9 | 86.8 | 43.4 | 83.5 | 26.4 |
| 10 | BeMapNet [50] | 67.7 | 56.7 | 36.5 | 33.1 | 71.9 | 75.4 | 87.2 | 43.1 | 82.5 | 23.6 |
| 11 | MapTR [41] | 49.3 | 41.1 | 29.3 | 28.5 | 55.4 | 53.7 | 62.6 | 30.7 | 60.3 | 8.2 |
| 13 | MapTR [41] | 49.7 | 38.1 | 26.0 | 26.2 | 53.9 | 57.3 | 55.4 | 23.5 | 54.9 | 7.6 |
| 14 | MapTR [41] | 50.4 | 38.7 | 22.6 | 24.0 | 54.7 | 55.8 | 59.3 | 27.0 | 60.0 | 5.8 |
| 15 | MapTR [41] | 58.7 | 49.3 | 34.4 | 31.8 | 71.2 | 46.2 | 90.9 | 33.3 | 83.0 | 3.7 |
| 16 | MapTR [41] | 48.7 | 49.3 | 26.0 | 25.7 | 71.1 | 48.8 | 88.4 | 48.1 | 79.0 | 7.1 |
| 17 | MapTR [41] | 50.1 | 48.1 | 28.4 | 27.9 | 69.2 | 42.6 | 88.4 | 47.4 | 77.0 | 4.1 |
| 18 | MapTRv2 [42] | 61.5 | 51.4 | 30.6 | 29.7 | 73.8 | 50.5 | 89.4 | 52.6 | 82.5 | 1.7 |
| 19 | StreamMapNet [73] | 63.4 | 54.4 | 21.2 | 24.4 | 75.8 | 69.9 | 90.0 | 57.0 | 82.6 | 14.3 |
| 20 | StreamMapNet [73] | 23.8 | 47.1 | 21.0 | 24.1 | 71.4 | 50.7 | 78.0 | 46.9 | 71.1 | 13.4 |
| 21 | StreamMapNet [73] | 28.0 | 55.5 | 36.7 | 43.1 | 79.5 | 62.9 | 83.3 | 51.3 | 79.8 | 71.0 |
| 22 | HIMap [77] | 65.5 | 56.6 | 29.7 | 29.0 | 79.4 | 64.9 | 93.0 | 62.0 | 87.2 | 7.8 |

- **Simulation *vs.* Real-World Data:** The corruptions applied in our benchmark are simulated to replicate real-world conditions. However, there may be discrepancies between simulated corruptions and actual real-world sensor failures or adverse weather conditions, which could affect the applicability of our results in real-world settings.

- **Model and Dataset Diversity:** Our benchmark includes 31 state-of-the-art HD map constructors, but it may not encompass the full diversity of available models and datasets. Future work could expand the benchmark to include more varied models and datasets to provide a more comprehensive evaluation.

- **Temporal and Spatial Consistency:** The benchmark focuses on the performance of models under specific corruptions applied at individual frames. Evaluating the temporal and spatial consistency of models under continuous adverse conditions remains an open challenge that is not fully addressed in this work.

Table 16: The CE (Corruption Error) of **LiDAR-only** HD map models in MapBench.

| # | Method | mAP | mCE | Fog | Wet | Snow | Motion | Beam | Crosstalk | Echo | Sensor |
|---|--------|-----|-----|-----|-----|------|--------|------|-----------|------|--------|
| 24 | MapTR [41] | 33.4 | 100.0 | 100.0 | 100.0 | 100.0 | 100.0 | 100.00 | 100.0 | 100.0 | 100.0 |
| 25 | MapTR [41] | 33.9 | 98.9 | 97.2 | 100.3 | 96.4 | 98.9 | 100.5 | 96.1 | 102.1 | 99.5 |
| 26 | MapTR [41] | 35.0 | 94.0 | 93.7 | 97.1 | 97.7 | 75.7 | 97.0 | 97.9 | 93.8 | 99.3 |
| 27 | MapTR [41] | 36.4 | 93.5 | 99.1 | 92.5 | 100.1 | 87.6 | 91.3 | 92.0 | 88.5 | 97.1 |
| 23 | VectorMapNet [43] | 31.6 | 94.9 | 115.9 | 95.8 | 80.4 | 93.5 | 90.8 | 88.3 | 104.3 | 90.3 |
| 28 | MapTRv2 [42] | 45.3 | 74.6 | 69.7 | 65.9 | 97.6 | 64.8 | 64.1 | 102.8 | 54.5 | 77.2 |
| 29 | HIMap [77] | 44.3 | 73.1 | 75.7 | 80.3 | 79.8 | 63.2 | 73.5 | 66.6 | 59.5 | 86.2 |

Table 17: The RR (Resilience Rate) of **LiDAR-only** HD map models in MapBench.

| # | Method | mAP | mRR | Fog | Wet | Snow | Motion | Beam | Crosstalk | Echo | Sensor |
|---|--------|-----|-----|-----|-----|------|--------|------|-----------|------|--------|
| 24 | MapTR [41] | 33.4 | 55.1 | 59.6 | 57.1 | 28.6 | 81.1 | 49.5 | 48.8 | 96.7 | 19.1 |
| 25 | MapTR [41] | 33.9 | 56.9 | 62.9 | 57.8 | 32.4 | 83.2 | 49.8 | 52.8 | 96.8 | 19.8 |
| 26 | MapTR [41] | 35.0 | 57.0 | 61.5 | 56.7 | 29.3 | 96.0 | 49.5 | 47.9 | 96.4 | 18.8 |
| 27 | MapTR [41] | 36.4 | 55.2 | 55.0 | 58.0 | 26.2 | 83.1 | 52.1 | 51.3 | 96.2 | 20.0 |
| 23 | VectorMapNet [43] | 31.6 | 63.4 | 49.6 | 64.1 | 50.3 | 91.0 | 60.9 | 62.4 | 99.2 | 30.0 |
| 28 | MapTRv2 [42] | 45.3 | 57.2 | 63.0 | 65.0 | 22.7 | 81.4 | 61.5 | 34.0 | 98.7 | 30.9 |
| 29 | HIMap [77] | 44.3 | 59.2 | 60.0 | 55.6 | 36.3 | 84.4 | 55.2 | 60.2 | 97.2 | 24.4 |

- **Computation and Resource Requirements:** Running extensive benchmarks on multiple models and corruption types is computationally intensive and resource-demanding. This limitation may restrict the accessibility of the benchmark to research groups with significant computational resources.

## G.2 Potential Negative Societal Impact

While the development of robust HD map construction methods has the potential to significantly advance autonomous driving technology, there are potential negative societal impacts that must be considered:

- **Privacy Concerns:** HD maps rely on detailed environmental data, which may include sensitive information about individuals and private properties. Ensuring the privacy and security of collected data is crucial to prevent misuse and protect individuals' rights.

- **Safety Risks:** While our benchmark aims to enhance the robustness of HD map models, there is a risk that reliance on these models could lead to overconfidence in autonomous systems. Ensuring that these systems are deployed with appropriate safety measures and human oversight is critical to prevent accidents and ensure public safety.

- **Environmental Impact:** The computational resources required to train and evaluate HD map models have a significant environmental footprint. Promoting the use of energy-efficient algorithms and sustainable computing practices is important to mitigate the environmental impact of this research.

- **Bias and Fairness:** The performance of HD map models may vary across different environments and conditions, potentially leading to biases in autonomous driving systems. Ensuring that these models are trained and evaluated on diverse datasets is crucial to promote fairness and prevent discriminatory outcomes.

## H Datasheets

In this section, we follow the NeurIPS Dataset and Benchmark guideline and use the template from Gebru *et al.* [14] to document necessary information about the proposed datasets and benchmarks.

Table 18: Complete list of 31 HD map construction models evaluated in MapBench. Basic information of different models includes input modality, BEV Encoder, backbone, training epoch, and performance on the clean nuScenes validation set. "L" and "C" represent LiDAR and camera, respectively. "Effi-B0", "R50", "PP", and "Sec" are short for EfficientNet-B0 [52], ResNet50 [18], PointPillars [32] and SECOND [68], respectively. † denotes the result is reproduced with the released model. ‡ means that we modify the public code and obtain results with the model trained by ourselves. $ped.$, $div.$, and $bou.$ are short for pedestrian-crossing, divider, and boundary, respectively.

| # | Method | Modal | Encoder | Data Aug | Temp | Back | Epoch | $AP_{ped.}$ | $AP_{div.}$ | $AP_{bou.}$ | mAP |
|---|--------|-------|---------|----------|------|------|-------|------|------|------|-----|
| 1 | HDMapNet [35] | C | NVT | ✗ | ✗ | Effi-B0 | 30 | 14.4 | 21.7 | 33.0 | 23.0 |
| 2 | VectorMapNet [43] | C | IPM | ✗ | ✗ | R50 | 110 | 36.1 | 47.3 | 39.3 | 40.9 |
| 3 | PivotNet [11] | C | PersFormer | ✗ | ✗ | R50 | 30 | 53.8 | 58.8 | 59.6 | 57.4 |
| 4 | PivotNet [11] | C | PersFormer | ✗ | ✗ | Effi-B0 | 30 | 53.9 | 59.7 | 61.0 | 58.2 |
| 5 | PivotNet [11] | C | PersFormer | ✗ | ✗ | SwinT | 30 | 58.7 | 63.8 | 64.9 | 62.5 |
| 6 | PivotNet [11] | C | PersFormer | ✗ | ✗ | SwinT | 110 | 62.6 | 68.0 | 69.7 | 66.8 |
| 7 | BeMapNet [50] | C | IPM-PE | ✗ | ✗ | R50 | 30 | 57.7 | 62.3 | 59.4 | 59.8 |
| 8 | BeMapNet [50] | C | IPM-PE | ✗ | ✗ | Effi-B0 | 30 | 56.0 | 62.2 | 59.0 | 59.1 |
| 9 | BeMapNet [50] | C | IPM-PE | ✗ | ✗ | SwinT | 30 | 61.3 | 64.4 | 61.6 | 62.5 |
| 10 | BeMapNet [50] | C | IPM-PE | ✗ | ✗ | SwinT | 110 | 64.4 | 69.0 | 69.7 | 67.7 |
| 11 | MapTR‡ [41] | C | GKT | ✗ | ✗ | R50 | 24 | 45.6 | 50.1 | 52.3 | 49.3 |
| 12 | MapTR [41] | C | GKT | PhotoMetric | ✗ | R50 | 24 | 46.3 | 51.5 | 53.1 | 50.3 |
| 13 | MapTR‡ [41] | C | GKT | Rotate | ✗ | R50 | 24 | 44.6 | 50.5 | 54.0 | 49.7 |
| 14 | MapTR‡ [41] | C | GKT | Flip | ✗ | R50 | 24 | 44.7 | 53.0 | 53.4 | 50.4 |
| 15 | MapTR [41] | C | GKT | PhotoMetric | ✗ | R50 | 110 | 56.2 | 59.8 | 60.1 | 58.7 |
| 16 | MapTR† [41] | C | BEVFormer | PhotoMetric | ✗ | R50 | 24 | 43.7 | 49.8 | 52.6 | 48.7 |
| 17 | MapTR† [41] | C | BEVPool | PhotoMetric | ✗ | R50 | 24 | 44.9 | 51.9 | 53.5 | 50.1 |
| 18 | MapTRv2 [42] | C | BEVPool | PhotoMetric | ✗ | R50 | 24 | 59.8 | 62.4 | 62.4 | 61.5 |
| 19 | StreamMapNet [73] | C | BEVFormer | PhotoMetric | ✗ | R50 | 30 | 61.7 | 66.3 | 62.1 | 63.4 |
| 20 | StreamMapNet† [73] | C | BEVFormer | PhotoMetric | ✗ | R50 | 30 | 17.2 | 22.6 | 31.6 | 23.8 |
| 21 | StreamMapNet‡ [73] | C | BEVFormer | PhotoMetric | ✓ | R50 | 30 | 21.4 | 27.4 | 35.2 | 28.0 |
| 22 | HIMap [77] | C | BEVFormer | PhotoMetric | ✗ | R50 | 24 | 62.2 | 66.5 | 67.9 | 65.5 |
| 23 | VectorMapNet [43] | L | — | ✗ | ✗ | PP | 110 | 25.7 | 37.6 | 38.6 | 34.0 |
| 24 | MapTR [41] | L | — | ✗ | ✗ | Sec | 24 | 48.5 | 53.7 | 64.7 | 55.6 |
| 25 | MapTR‡ [41] | L | — | Dropout | ✗ | Sec | 24 | 49.5 | 55.3 | 66.4 | 57.0 |
| 26 | MapTR‡ [41] | L | — | RTS-LiDAR | ✗ | Sec | 24 | 48.7 | 56.2 | 66.9 | 57.3 |
| 27 | MapTR‡ [41] | L | — | PolarMix | ✗ | Sec | 24 | 53.7 | 57.5 | 69.5 | 60.2 |
| 28 | MapTRv2 [42] | L | — | ✗ | ✗ | Sec | 24 | 56.6 | 58.1 | 69.8 | 61.5 |
| 29 | HIMap‡ [77] | L | — | ✗ | ✗ | Sec | 24 | 54.8 | 64.7 | 73.5 | 64.3 |
| 30 | MapTR [41] | C & L | GKT | PhotoMetric | ✗ | R50 & Sec | 24 | 55.9 | 62.3 | 69.3 | 62.5 |
| 31 | HIMap [77] | C & L | BEVFormer | PhotoMetric | ✗ | R50 & Sec | 24 | 71.0 | 72.4 | 79.4 | 74.3 |

## H.1 Motivation

The questions in this section are primarily intended to encourage dataset creators to clearly articulate their reasons for creating the dataset and to promote transparency about funding interests. The latter may be particularly relevant for datasets created for research purposes.

1. *"For what purpose was the dataset created?"*

   **A:** The dataset was created to facilitate relevant research in the area of HD map construction robustness under out-of-distribution sensor corruptions.

2. *"Who created the dataset (e.g., which team, research group) and on behalf of which entity?"*

   **A:** The dataset was created by:
   - Xiaoshuai Hao (Samsung R&D Institute China–Beijing),
   - Mengchuan Wei (Samsung R&D Institute China–Beijing),
   - Yifan Yang (Samsung R&D Institute China–Beijing),
   - Haimei Zhao (The University of Sydney),
   - Hui Zhang (Samsung R&D Institute China–Beijing),
   - Yi Zhou (Samsung R&D Institute China–Beijing),
   - Qiang Wang (Samsung R&D Institute China–Beijing),
   - Weiming Li (Samsung R&D Institute China–Beijing),
   - Lingdong Kong (National University of Singapore),
   - Jing Zhang (The University of Sydney).

3. *"Who funded the creation of the dataset?"*

   **A:** The creation of the dataset is funded by related affiliations of the authors in this work, *i.e.*, Samsung R&D Institute China–Beijing, the National University of Singapore, and the University of Sydney.

Table 19: The `mAP` metrics of different **camera-only** HD map models in MapBench.

| # | Method | Clean | Camera | Frame | Quant | Motion | Bright | Dark | Fog | Snow |
|---|--------|-------|--------|-------|-------|--------|--------|------|-----|------|
| 1 | HDMapNet [35] | 23.0 | 4.6 | 5.1 | 18.9 | 20.8 | 16.7 | 9.3 | 10.6 | 5.2 |
| 2 | VectorMapNet [43] | 40.9 | 13.9 | 12.3 | 26.6 | 29.7 | 25.2 | 7.8 | 18.3 | 2.9 |
| 3 | PivotNet [11] | 57.4 | 17.1 | 16.7 | 36.4 | 34.1 | 43.5 | 16.5 | 37.4 | 4.6 |
| 4 | PivotNet [11] | 58.2 | 16.6 | 16.2 | 40.7 | 38.4 | 47.5 | 19.1 | 43.7 | 8.6 |
| 5 | PivotNet [11] | 62.5 | 17.8 | 18.0 | 42.8 | 41.0 | 51.7 | 22.6 | 49.5 | 7.5 |
| 6 | PivotNet [11] | 66.8 | 20.2 | 20.0 | 43.4 | 45.2 | 54.8 | 21.8 | 52.9 | 6.8 |
| 7 | BeMapNet [50] | 59.8 | 18.8 | 18.5 | 38.1 | 35.3 | 50.7 | 23.2 | 46.5 | 9.6 |
| 8 | BeMapNet [50] | 59.1 | 18.2 | 17.6 | 39.7 | 42.0 | 50.7 | 28.7 | 44.2 | 13.7 |
| 9 | BeMapNet [50] | 62.5 | 22.9 | 20.7 | 48.0 | 47.4 | 54.2 | 27.1 | 52.1 | 16.5 |
| 10 | BeMapNet [50] | 67.7 | 24.5 | 22.2 | 48.2 | 50.5 | 58.4 | 28.9 | 55.3 | 15.9 |
| 11 | MapTR‡ [41] | 49.3 | 14.5 | 14.1 | 27.3 | 26.5 | 30.9 | 15.1 | 29.7 | 4.0 |
| 12 | MapTR [41] | 50.3 | 15.0 | 14.2 | 35.4 | 23.5 | 44.3 | 22.7 | 38.5 | 3.8 |
| 13 | MapTR‡ [41] | 49.7 | 12.9 | 13.0 | 26.8 | 28.5 | 27.5 | 11.7 | 27.3 | 3.8 |
| 14 | MapTR‡ [41] | 50.4 | 11.4 | 12.1 | 27.6 | 28.1 | 29.9 | 13.6 | 30.2 | 2.9 |
| 15 | MapTR [41] | 58.7 | 20.4 | 18.9 | 42.3 | 27.4 | 53.9 | 19.7 | 49.2 | 2.2 |
| 16 | MapTR† [41] | 48.7 | 12.7 | 12.5 | 34.6 | 23.8 | 43.1 | 23.4 | 38.5 | 3.4 |
| 17 | MapTR† [41] | 50.1 | 14.2 | 14.0 | 34.7 | 21.3 | 44.3 | 23.7 | 38.6 | 2.0 |
| 18 | MapTRv2 [42] | 61.5 | 18.8 | 18.2 | 45.3 | 31.0 | 54.9 | 32.3 | 50.7 | 1.1 |
| 19 | StreamMapNet [73] | 63.4 | 13.4 | 15.5 | 48.1 | 44.3 | 57.0 | 36.1 | 52.4 | 9.1 |
| 20 | StreamMapNet† [73] | 23.8 | 5.0 | 5.7 | 17.0 | 12.1 | 18.6 | 11.2 | 16.9 | 3.2 |
| 21 | StreamMapNet‡ [73] | 28.0 | 10.3 | 12.1 | 22.3 | 17.6 | 23.3 | 14.4 | 22.3 | 2.0 |
| 22 | HIMap‡ [77] | 65.5 | 19.4 | 19.0 | 52.0 | 42.5 | 60.9 | 40.6 | 57.1 | 5.1 |

## H.2 Composition

Most of the questions in this section are intended to provide dataset consumers with the information they need to make informed decisions about using the dataset for their chosen tasks. Some of the questions are designed to elicit information about compliance with the EU's General Data Protection Regulation (GDPR) or comparable regulations in other jurisdictions. Questions that apply only to datasets that relate to people are grouped together at the end of the section. We recommend taking a broad interpretation of whether a dataset relates to people. For example, any dataset containing text that was written by people relates to people.

1. *"What do the instances that comprise our datasets represent (e.g., documents, photos, people, countries)?"*

   **A:** The instances that comprise the dataset are mainly images and LiDAR point clouds captured by the camera and LiDAR sensors, respectively, providing visual representations of outdoor driving scenes observed.

2. *"How many instances are there in total (of each type, if appropriate)?"*

   **A:** The dataset contains a total of 29 types of corruptions that occur from cameras and LiDAR sensors, including 8 types of camera corruptions, 8 types of LiDAR corruptions, and 13 types of multi-sensor corruptions.

3. *"Does the dataset contain all possible instances or is it a sample (not necessarily random) of instances from a larger set?"*

   **A:** Yes, our dataset contains all possible instances that have been collected so far.

4. *"Is there a label or target associated with each instance?"*

   **A:** Yes, each instance in our dataset is associated with a label for either the RGB image or LiDAR point cloud.

5. *"Is any information missing from individual instances?"*

   **A:** No.

6. *"Are relationships between individual instances made explicit (e.g., users' movie ratings, social network links)?"*

   **A:** Yes, the relationship between individual instances is explicit.

Table 20: The $\text{AP}_{ped.}$ metric of different **camera-only** HD map models in MapBench.

| # | Method | Clean | Camera | Frame | Quant | Motion | Bright | Dark | Fog | Snow |
|---|--------|-------|--------|-------|-------|--------|--------|------|-----|------|
| 1 | HDMapNet [35] | 14.4 | 9.5 | 6.9 | 15.2 | 15.8 | 12.7 | 8.2 | 8.1 | 3.7 |
| 2 | VectorMapNet [43] | 36.1 | 16.6 | 12.6 | 24.2 | 27.1 | 23.2 | 6.6 | 16.6 | 0.9 |
| 3 | PivotNet [11] | 53.8 | 24.2 | 19.2 | 33.9 | 31.9 | 40.8 | 12.5 | 35.0 | 1.8 |
| 4 | PivotNet [11] | 53.9 | 23.0 | 17.8 | 38.4 | 35.3 | 44.6 | 14.2 | 41.8 | 2.7 |
| 5 | PivotNet [11] | 58.7 | 25.2 | 20.3 | 40.0 | 36.0 | 48.9 | 18.9 | 47.3 | 2.2 |
| 6 | PivotNet [11] | 62.6 | 29.0 | 23.6 | 40.6 | 39.9 | 51.8 | 18.2 | 51.2 | 3.3 |
| 7 | BeMapNet [50] | 57.7 | 24.5 | 21.1 | 35.7 | 31.7 | 47.7 | 17.8 | 43.9 | 5.4 |
| 8 | BeMapNet [50] | 56.0 | 22.8 | 19.0 | 37.0 | 36.9 | 47.3 | 23.7 | 40.1 | 4.4 |
| 9 | BeMapNet [50] | 61.3 | 26.5 | 22.5 | 44.8 | 43.6 | 52.6 | 23.8 | 50.9 | 12.3 |
| 10 | BeMapNet [50] | 64.4 | 29.0 | 24.2 | 44.6 | 46.0 | 55.9 | 24.8 | 53.3 | 10.1 |
| 11 | MapTR‡ [41] | 45.6 | 18.4 | 14.7 | 22.7 | 24.6 | 26.3 | 12.2 | 25.8 | 2.2 |
| 12 | MapTR [41] | 46.3 | 18.0 | 13.9 | 31.2 | 22.0 | 39.6 | 17.8 | 34.8 | 2.0 |
| 13 | MapTR‡ [41] | 44.6 | 17.4 | 13.6 | 22.1 | 26.7 | 23.3 | 7.2 | 24.1 | 3.0 |
| 14 | MapTR‡ [41] | 44.7 | 16.9 | 12.4 | 22.8 | 26.5 | 24.2 | 9.0 | 26.0 | 2.3 |
| 15 | MapTR [41] | 56.2 | 22.9 | 18.8 | 38.4 | 26.6 | 49.8 | 14.4 | 46.3 | 0.5 |
| 16 | MapTR† [41] | 43.7 | 17.3 | 13.4 | 30.6 | 24.2 | 39.1 | 19.4 | 35.1 | 2.7 |
| 17 | MapTR† [41] | 44.9 | 18.3 | 14.0 | 29.3 | 20.1 | 38.9 | 18.5 | 33.4 | 1.2 |
| 18 | MapTRv2 [42] | 59.8 | 25.8 | 20.0 | 43.8 | 31.0 | 53.5 | 29.4 | 49.7 | 0.6 |
| 19 | StreamMapNet [73] | 61.7 | 20.6 | 17.7 | 46.3 | 42.8 | 55.7 | 33.6 | 51.5 | 6.9 |
| 20 | StreamMapNet† [73] | 17.2 | 4.4 | 4.0 | 11.8 | 6.8 | 11.3 | 5.4 | 12.9 | 0.8 |
| 21 | StreamMapNet‡ [73] | 21.4 | 8.3 | 10.3 | 17.7 | 12.8 | 15.7 | 10.9 | 18.7 | 0.9 |
| 22 | HIMap‡ [77] | 62.2 | 27.3 | 21.7 | 48.6 | 41.2 | 57.4 | 36.4 | 53.3 | 2.6 |

7. *"Are there recommended data splits (e.g., training, development/validation, testing)?"*

   **A:** Yes, we provide detailed data splits for our dataset.

8. *"Is the dataset self-contained, or does it link to or otherwise rely on external resources (e.g., websites, tweets, other datasets)?"*

   **A:** Yes, our datasets are self-contained.

9. *"Does the dataset contain data that might be considered confidential (e.g., data that is protected by legal privilege or by doctor–patient confidentiality, data that includes the content of individuals' non-public communications)?"*

   **A:** No, all data are clearly licensed.

10. *"Does the dataset contain data that, if viewed directly, might be offensive, insulting, threatening, or might otherwise cause anxiety?"*

    **A:** No.

## H.3 Collection Process

In addition to the goals outlined in the previous section, the questions in this section are designed to elicit information that may help researchers and practitioners create alternative datasets with similar characteristics. Again, questions that apply only to datasets that relate to people are grouped together at the end of the section.

1. *"How was the data associated with each instance acquired?"*

   **A:** Please refer to the details listed in Sec. A.

2. *"What mechanisms or procedures were used to collect the data (e.g., hardware apparatuses or sensors, manual human curation, software programs, software APIs)?"*

   **A:** Please refer to the details listed in Sec. A.

3. *"If the dataset is a sample from a larger set, what was the sampling strategy (e.g., deterministic, probabilistic with specific sampling probabilities)?"*

   **A:** Please refer to the details listed in Sec. A.

Table 21: The $AP_{div.}$ metrics of different **camera-only** HD map models in MapBench.

| # | Model | Clean | Camera | Frame | Quant | Motion | Bright | Dark | Fog | Snow |
|---|---|---|---|---|---|---|---|---|---|---|
| 1 | HDMapNet [35] | 21.7 | 3.4 | 4.6 | 16.4 | 17.7 | 14.0 | 6.5 | 8.9 | 4.2 |
| 2 | VectorMapNet [43] | 47.3 | 17.3 | 15.3 | 30.4 | 34.4 | 29.0 | 8.8 | 22.2 | 3.7 |
| 3 | PivotNet [11] | 58.8 | 15.8 | 16.2 | 35.5 | 33.4 | 42.6 | 19.0 | 37.5 | 5.8 |
| 4 | PivotNet [11] | 59.7 | 17.5 | 17.0 | 40.0 | 38.7 | 47.0 | 21.7 | 43.3 | 10.8 |
| 5 | PivotNet [11] | 63.8 | 17.7 | 18.1 | 42.4 | 42.4 | 52.4 | 24.4 | 49.6 | 8.6 |
| 6 | PivotNet [11] | 68.0 | 19.7 | 20.0 | 43.5 | 47.1 | 55.6 | 24.0 | 53.4 | 8.2 |
| 7 | BeMapNet [50] | 62.3 | 18.1 | 18.2 | 38.4 | 36.5 | 52.9 | 26.3 | 49.1 | 9.6 |
| 8 | BeMapNet [50] | 62.2 | 16.6 | 16.7 | 40.3 | 44.3 | 53.3 | 30.8 | 46.1 | 16.7 |
| 9 | BeMapNet [50] | 64.4 | 21.6 | 20.1 | 50.1 | 49.8 | 55.8 | 29.2 | 54.0 | 17.6 |
| 10 | BeMapNet [50] | 69.0 | 22.5 | 21.6 | 50.1 | 53.3 | 59.8 | 31.9 | 56.6 | 18.4 |
| 11 | MapTR‡ [41] | 50.1 | 15.7 | 15.4 | 28.3 | 25.8 | 31.8 | 15.8 | 31.1 | 4.6 |
| 12 | MapTR [41] | 51.5 | 14.5 | 15.1 | 36.5 | 23.2 | 46.2 | 24.4 | 39.9 | 4.7 |
| 13 | MapTR‡ [41] | 50.5 | 12.8 | 13.9 | 27.1 | 28.3 | 26.8 | 13.0 | 28.3 | 4.0 |
| 14 | MapTR‡ [41] | 53.0 | 11.1 | 13.8 | 28.9 | 28.2 | 31.4 | 16.2 | 32.0 | 3.3 |
| 15 | MapTR [41] | 59.8 | 21.1 | 20.1 | 43.5 | 27.1 | 55.6 | 21.8 | 51.9 | 2.5 |
| 16 | MapTR† [41] | 49.8 | 12.5 | 13.1 | 34.8 | 23.4 | 43.1 | 25.1 | 39.6 | 3.5 |
| 17 | MapTR† [41] | 51.9 | 14.5 | 15.2 | 36.1 | 22.1 | 45.9 | 26.2 | 40.9 | 1.6 |
| 18 | MapTRv2 [42] | 62.4 | 18.1 | 18.6 | 45.1 | 30.6 | 54.7 | 33.1 | 50.5 | 1.0 |
| 19 | StreamMapNet [73] | 66.3 | 14.1 | 16.9 | 50.5 | 46.7 | 60.2 | 39.6 | 55.7 | 9.7 |
| 20 | StreamMapNet† [73] | 22.6 | 5.7 | 6.1 | 16.3 | 13.3 | 17.9 | 12.4 | 16.8 | 3.1 |
| 21 | StreamMapNet‡ [73] | 27.4 | 10.3 | 11.4 | 21.5 | 18.4 | 23.6 | 15.0 | 22.4 | 1.9 |
| 22 | HIMap‡ [77] | 66.5 | 19.4 | 19.1 | 53.0 | 43.1 | 62.2 | 42.9 | 59.7 | 6.1 |

## H.4 Preprocessing, Cleaning, and Labeling

The questions in this section are intended to provide dataset consumers with the information they need to determine whether the "raw" data has been processed in ways that are compatible with their chosen tasks. For example, text that has been converted into a "bag-of-words" is not suitable for tasks involving word order.

1. *"Was any preprocessing/cleaning/labeling of the data done (e.g., discretization or bucketing, tokenization, part-of-speech tagging, SIFT feature extraction, removal of instances, processing of missing values)?"*

   **A:** Yes, we preprocessed and cleaned data in our dataset.

2. *"Was the 'raw' data saved in addition to the preprocessed/cleaned/labeled data (e.g., to support unanticipated future uses)?"*

   **A:** Yes, raw data is accessible.

3. *"Is the software that was used to preprocess/clean/label the data available?"*

   **A:** Yes, the necessary software used to preprocess and clean the data is publicly available.

## H.5 Uses

The questions in this section are intended to encourage dataset creators to reflect on tasks for which the dataset should and should not be used. By explicitly highlighting these tasks, dataset creators can help dataset consumers make informed decisions, thereby avoiding potential risks or harms.

1. *"Has the dataset been used for any tasks already?"*

   **A:** No.

2. *"Is there a repository that links to any or all papers or systems that use the dataset?"*

   **A:** Yes, we provide such links in our GitHub repository.

3. *"What (other) tasks could the dataset be used for?"*

   **A:** The dataset could be used for relevant perception, tracking, and planning tasks based on camera and LiDAR sensors.

Table 22: The $\text{AP}_{bou.}$ metric of different **camera-only** HD map models in MapBench.

| # | Method | Clean | Camera | Frame | Quant | Motion | Bright | Dark | Fog | Snow |
|---|--------|-------|--------|-------|-------|--------|--------|------|-----|------|
| 1 | HDMapNet [35] | 33.0 | 0.8 | 3.8 | 24.9 | 28.8 | 23.6 | 13.1 | 14.8 | 7.8 |
| 2 | VectorMapNet [43] | 39.3 | 8.0 | 9.1 | 25.1 | 27.7 | 23.5 | 8.1 | 16.1 | 4.0 |
| 3 | PivotNet [11] | 59.6 | 11.2 | 14.6 | 39.7 | 37.1 | 47.0 | 18.1 | 39.7 | 6.1 |
| 4 | PivotNet [11] | 61.0 | 9.4 | 13.8 | 43.9 | 41.1 | 50.8 | 21.4 | 46.2 | 12.4 |
| 5 | PivotNet [11] | 64.9 | 10.6 | 15.5 | 46.0 | 44.8 | 53.8 | 24.5 | 51.5 | 11.7 |
| 6 | PivotNet [11] | 69.7 | 11.7 | 16.5 | 46.1 | 48.6 | 56.9 | 23.2 | 54.0 | 8.9 |
| 7 | BeMapNet [50] | 59.4 | 13.6 | 16.2 | 40.3 | 37.8 | 51.5 | 25.4 | 46.6 | 13.7 |
| 8 | BeMapNet [50] | 59.0 | 15.0 | 16.9 | 41.9 | 44.9 | 51.4 | 31.6 | 46.3 | 19.9 |
| 9 | BeMapNet [50] | 61.6 | 20.6 | 19.5 | 49.2 | 48.6 | 54.1 | 28.3 | 51.4 | 19.5 |
| 10 | BeMapNet [50] | 69.7 | 22.0 | 20.8 | 49.8 | 52.3 | 59.4 | 30.0 | 56.0 | 19.1 |
| 11 | MapTR‡ [41] | 52.3 | 9.4 | 12.0 | 31.0 | 29.0 | 34.5 | 17.4 | 32.3 | 5.4 |
| 12 | MapTR [41] | 53.1 | 12.4 | 13.4 | 38.3 | 25.2 | 47.3 | 26.0 | 40.7 | 4.8 |
| 13 | MapTR‡ [41] | 54.0 | 8.6 | 11.5 | 31.2 | 30.4 | 32.5 | 14.8 | 29.4 | 4.3 |
| 14 | MapTR‡ [41] | 53.4 | 6.1 | 10.1 | 31.1 | 29.8 | 34.1 | 15.7 | 32.6 | 3.2 |
| 15 | MapTR [41] | 60.1 | 17.1 | 17.8 | 44.9 | 28.4 | 56.3 | 23.1 | 49.4 | 3.5 |
| 16 | MapTR† [41] | 52.6 | 8.3 | 11.0 | 38.5 | 23.7 | 47.0 | 25.8 | 40.8 | 4.1 |
| 17 | MapTR† [41] | 53.5 | 10.0 | 12.8 | 38.6 | 21.8 | 48.0 | 26.5 | 41.5 | 3.3 |
| 18 | MapTRv2 [42] | 62.4 | 12.5 | 16.1 | 47.1 | 31.4 | 56.6 | 34.5 | 51.8 | 1.6 |
| 19 | StreamMapNet [73] | 62.1 | 5.5 | 11.9 | 47.4 | 43.5 | 55.3 | 35.2 | 49.9 | 10.5 |
| 20 | StreamMapNet† [73] | 31.6 | 4.9 | 7.0 | 22.8 | 16.0 | 26.5 | 15.7 | 21.2 | 5.6 |
| 21 | StreamMapNet‡ [73] | 35.2 | 12.2 | 14.5 | 27.7 | 21.7 | 30.7 | 17.3 | 25.9 | 3.2 |
| 22 | HIMap‡ [77] | 67.9 | 11.6 | 16.1 | 54.5 | 43.3 | 63.2 | 42.4 | 58.4 | 6.6 |

4. *"Is there anything about the composition of the dataset or the way it was collected and preprocessed/cleaned/labeled that might impact future uses?"*

   A: N/A.

5. *"Are there tasks for which the dataset should not be used?"*

   A: N/A.

## H.6 Distribution

Dataset creators should provide answers to these questions prior to distributing the dataset either internally within the entity on behalf of which the dataset was created or externally to third parties.

1. *"Will the dataset be distributed to third parties outside of the entity (e.g., company, institution, organization) on behalf of which the dataset was created?"*

   A: No.

2. *"How will the dataset be distributed (e.g., tarball on website, API, GitHub)?"*

   A: Very likely to be distributed by website, API, and GitHub repository.

3. *"When will the dataset be distributed?"*

   A: The datasets are publicly accessible.

4. *"Will the dataset be distributed under a copyright or other intellectual property (IP) license, and/or under applicable terms of use (ToU)?"*

   A: Yes, the dataset is under the Creative Commons Attribution-NonCommercial-ShareAlike 4.0 International License.

5. *"Have any third parties imposed IP-based or other restrictions on the data associated with the instances?"*

   A: No.

6. *"Do any export controls or other regulatory restrictions apply to the dataset or to individual instances?"*

   A: No.

Table 23: The $\mathtt{AP}_{ped.}$, $\mathtt{AP}_{div.}$, $\mathtt{AP}_{bou.}$, and $\mathtt{mAP}$ metrics of **LiDAR-only** models in MapBench.

| # | Method | Metric | Clean | Fog | Wet | Snow | Motion | Beam | Crosstalk | Echo | Sensor |
|---|--------|--------|-------|-----|-----|------|--------|------|-----------|------|--------|
| 23 | VectorMapNet‡ [43] | $\mathtt{AP}_{ped.}$ | 26.8 | 13.3 | 14.6 | 10.9 | 24.8 | 14.9 | 15.2 | 27.2 | 6.6 |
| 23 | VectorMapNet‡ [43] | $\mathtt{AP}_{div.}$ | 32.5 | 16.3 | 22.1 | 17.0 | 29.6 | 21.2 | 20.5 | 31.9 | 10.9 |
| 23 | VectorMapNet‡ [43] | $\mathtt{AP}_{bou.}$ | 35.4 | 17.4 | 24.1 | 19.9 | 32.0 | 21.7 | 23.5 | 34.8 | 10.8 |
| 23 | VectorMapNet‡ [43] | $\mathtt{mAP}$ | 31.6 | 15.7 | 20.3 | 15.9 | 28.8 | 19.2 | 19.7 | 31.3 | 9.5 |
| 24 | MapTR‡ [41] | $\mathtt{AP}_{ped.}$ | 26.6 | 15.8 | 13.4 | 5.7 | 21.7 | 11.3 | 13.6 | 26.3 | 3.5 |
| 24 | MapTR‡ [41] | $\mathtt{AP}_{div.}$ | 31.7 | 19.4 | 17.8 | 8.3 | 25.4 | 15.7 | 15.0 | 29.9 | 6.6 |
| 24 | MapTR‡ [41] | $\mathtt{AP}_{bou.}$ | 41.8 | 24.5 | 26.0 | 14.8 | 34.2 | 22.6 | 20.3 | 40.6 | 8.9 |
| 24 | MapTR‡ [41] | $\mathtt{mAP}$ | 33.4 | 19.9 | 19.1 | 9.6 | 27.1 | 16.5 | 16.3 | 32.3 | 6.4 |
| 25 | MapTR‡ [41] | $\mathtt{AP}_{ped.}$ | 27.4 | 17.6 | 13.8 | 6.8 | 22.9 | 12.6 | 12.8 | 27.1 | 4.9 |
| 25 | MapTR‡ [41] | $\mathtt{AP}_{div.}$ | 30.0 | 18.8 | 17.2 | 8.3 | 24.3 | 14.3 | 15.5 | 27.9 | 5.8 |
| 25 | MapTR‡ [41] | $\mathtt{AP}_{bou.}$ | 41.5 | 25.6 | 25.9 | 16.9 | 34.9 | 22.3 | 23.7 | 40.5 | 8.8 |
| 25 | MapTR‡ [41] | $\mathtt{mAP}$ | 33.9 | 20.7 | 19.0 | 10.7 | 27.4 | 16.4 | 17.4 | 31.8 | 6.5 |
| 26 | MapTR‡ [41] | $\mathtt{AP}_{ped.}$ | 28.3 | 17.7 | 14.1 | 6.4 | 27.1 | 12.5 | 13.6 | 27.6 | 4.4 |
| 26 | MapTR‡ [41] | $\mathtt{AP}_{div.}$ | 32.7 | 20.4 | 18.7 | 8.7 | 31.2 | 16.1 | 15.1 | 30.8 | 6.8 |
| 26 | MapTR‡ [41] | $\mathtt{AP}_{bou.}$ | 44.1 | 26.4 | 26.8 | 15.9 | 42.4 | 23.4 | 21.5 | 42.8 | 8.5 |
| 26 | MapTR‡ [41] | $\mathtt{mAP}$ | 35.0 | 21.5 | 19.8 | 10.3 | 33.6 | 17.3 | 16.8 | 33.7 | 6.6 |
| 27 | MapTR‡ [41] | $\mathtt{AP}_{ped.}$ | 30.1 | 17.9 | 16.0 | 6.3 | 25.7 | 14.0 | 15.9 | 29.9 | 4.4 |
| 27 | MapTR‡ [41] | $\mathtt{AP}_{div.}$ | 33.0 | 18.3 | 18.7 | 7.1 | 26.4 | 16.8 | 15.6 | 30.8 | 6.7 |
| 27 | MapTR‡ [41] | $\mathtt{AP}_{bou.}$ | 46.1 | 23.8 | 28.6 | 15.1 | 38.5 | 26.1 | 24.5 | 44.3 | 10.7 |
| 27 | MapTR‡ [41] | $\mathtt{mAP}$ | 36.4 | 20.0 | 21.1 | 9.5 | 30.2 | 19.0 | 18.7 | 35.0 | 7.3 |
| 28 | MapTRv2‡ [42] | $\mathtt{AP}_{ped.}$ | 38.5 | 23.9 | 21.7 | 6.4 | 31.9 | 21.4 | 11.4 | 39.0 | 9.3 |
| 28 | MapTRv2‡ [42] | $\mathtt{AP}_{div.}$ | 43.5 | 27.0 | 28.8 | 6.4 | 33.6 | 26.3 | 11.6 | 41.9 | 12.6 |
| 28 | MapTRv2‡ [42] | $\mathtt{AP}_{bou.}$ | 54.0 | 34.6 | 37.9 | 18.1 | 45.0 | 35.8 | 23.2 | 53.2 | 20.1 |
| 28 | MapTRv2‡ [42] | $\mathtt{mAP}$ | 45.3 | 28.5 | 29.5 | 10.3 | 36.9 | 27.9 | 15.4 | 44.7 | 14.0 |
| 29 | HIMap‡ [77] | $\mathtt{AP}_{ped.}$ | 35.8 | 23.0 | 16.6 | 8.5 | 31.4 | 18.6 | 20.9 | 35.4 | 7.8 |
| 29 | HIMap‡ [77] | $\mathtt{AP}_{div.}$ | 43.3 | 26.2 | 23.7 | 14.8 | 35.2 | 22.8 | 23.8 | 41.1 | 10.2 |
| 29 | HIMap‡ [77] | $\mathtt{AP}_{bou.}$ | 54.0 | 30.5 | 33.6 | 25.1 | 45.6 | 31.9 | 35.3 | 52.7 | 14.4 |
| 29 | HIMap‡ [77] | $\mathtt{mAP}$ | 44.3 | 26.6 | 24.6 | 16.1 | 37.4 | 24.4 | 26.7 | 43.1 | 10.8 |

## H.7 Maintenance

As with the questions in the previous section, dataset creators should provide answers to these questions prior to distributing the dataset. The questions in this section are intended to encourage dataset creators to plan for dataset maintenance and communicate this plan to dataset consumers.

1. *"Who will be supporting/hosting/maintaining the dataset?"*

   **A:** The authors of this work serve to support, host, and maintain the datasets.

2. *"How can the owner/curator/manager of the dataset be contacted (e.g., email address)?"*

   **A:** The curators can be contacted via the email addresses listed on our webpage[1].

3. *"Is there an erratum?"*

   **A:** There is no explicit erratum; updates and known errors will be specified in future versions.

4. *"Will the dataset be updated (e.g., to correct labeling errors, add new instances, delete instances)?"*

   **A:** No, for the current version. Future updates (if any) will be posted on the dataset website.

5. *"Will older versions of the dataset continue to be supported/hosted/maintained?"*

   **A:** Yes. This is the first version of the release; future updates will be posted and older versions will be replaced.

6. *"If others want to extend/augment/build on/contribute to the dataset, is there a mechanism for them to do so?"*

   **A:** Yes, we provide detailed instructions for future extensions.

---

[1] https://mapbench.github.io.

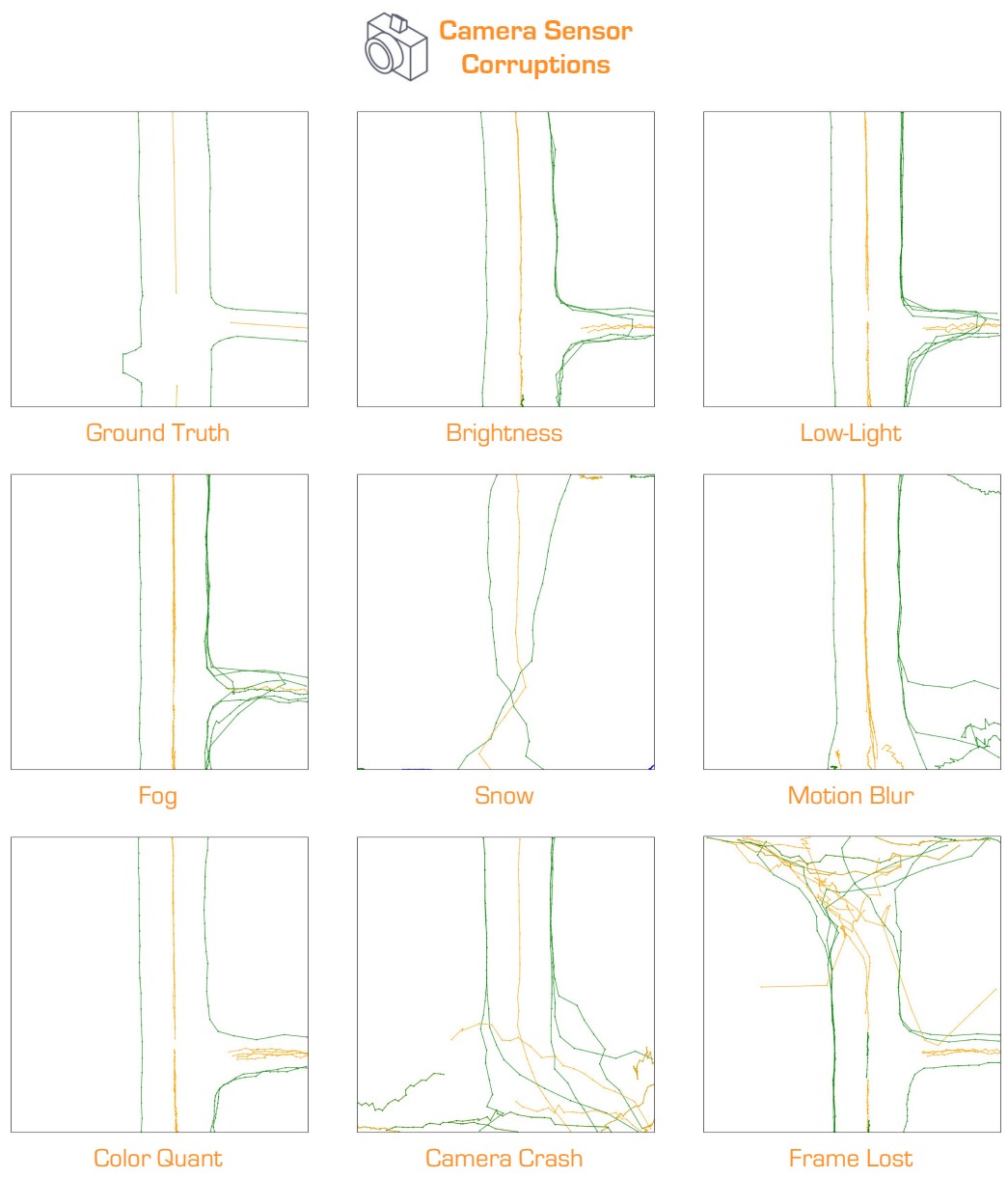

Figure 9: Qualitative assessment of camera-only HD map construction under the Camera sensor corruptions. Best viewed in color and zoomed in for details.

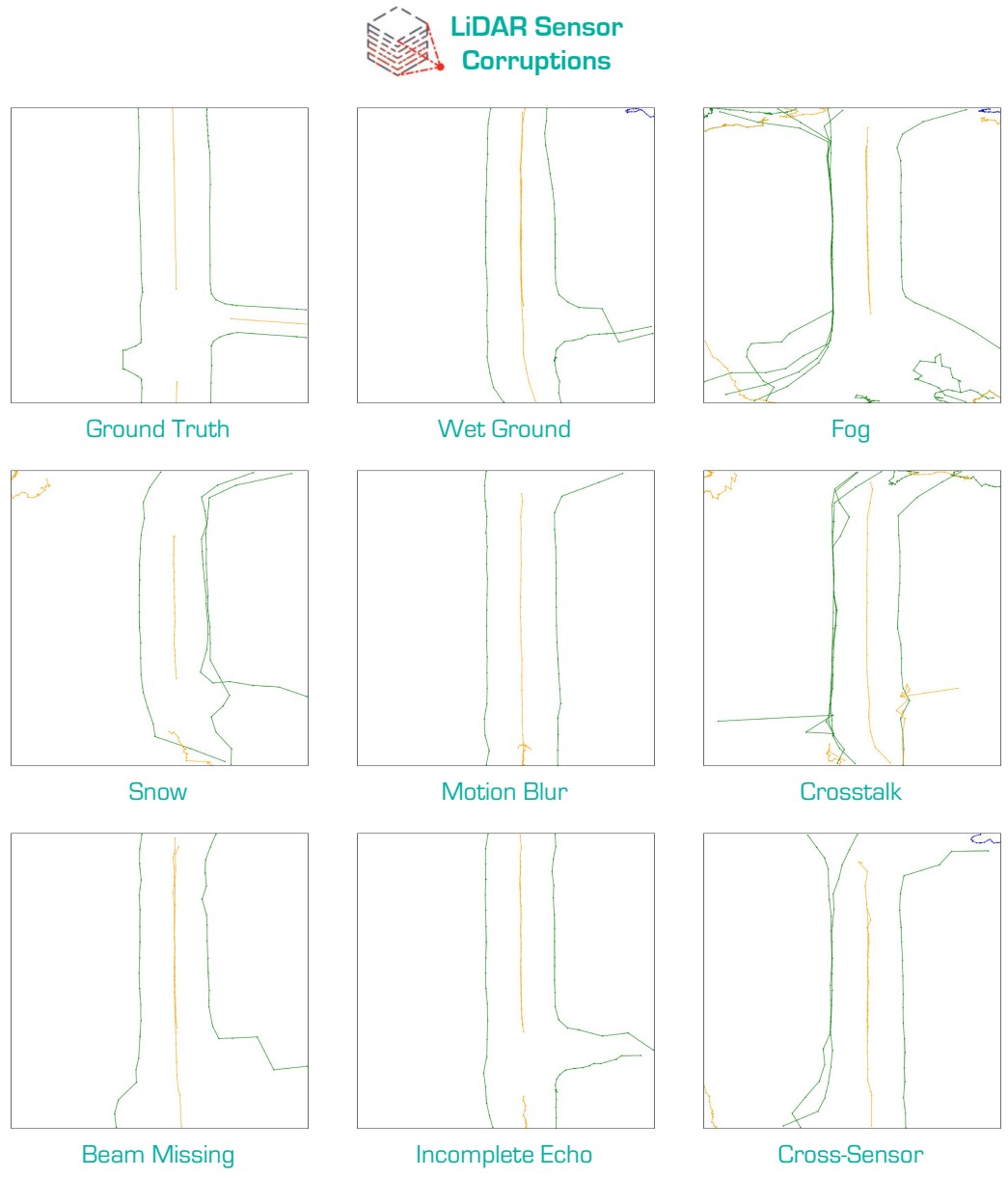

Figure 10: Qualitative assessment of LiDAR-only HD map construction under the LiDAR sensor corruptions. Best viewed in color and zoomed in for details.

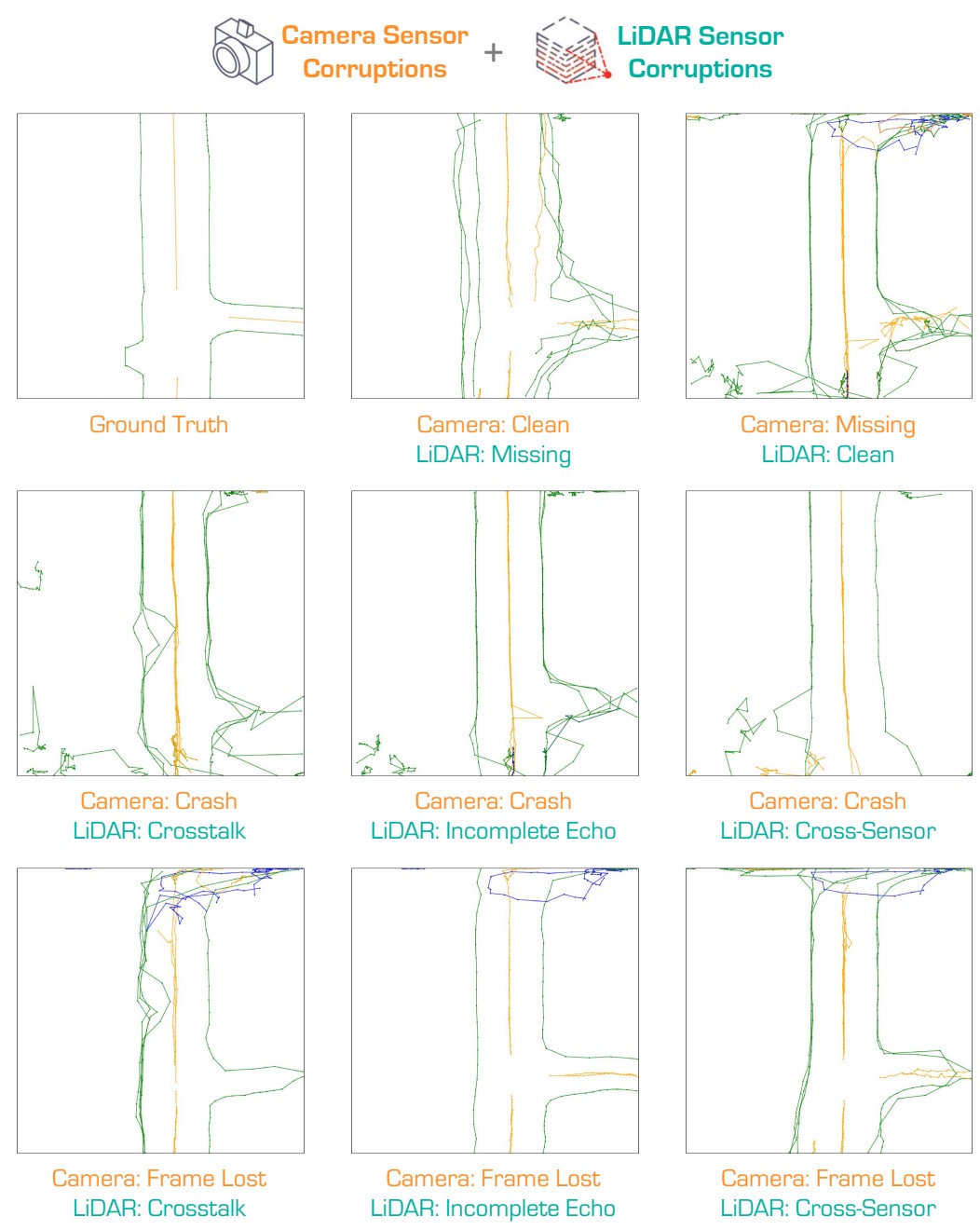

Figure 11: Qualitative assessment of camera-LiDAR fusion-based HD map construction under the Camera and LiDAR combined sensor corruptions. Best viewed in color and zoomed in for details.

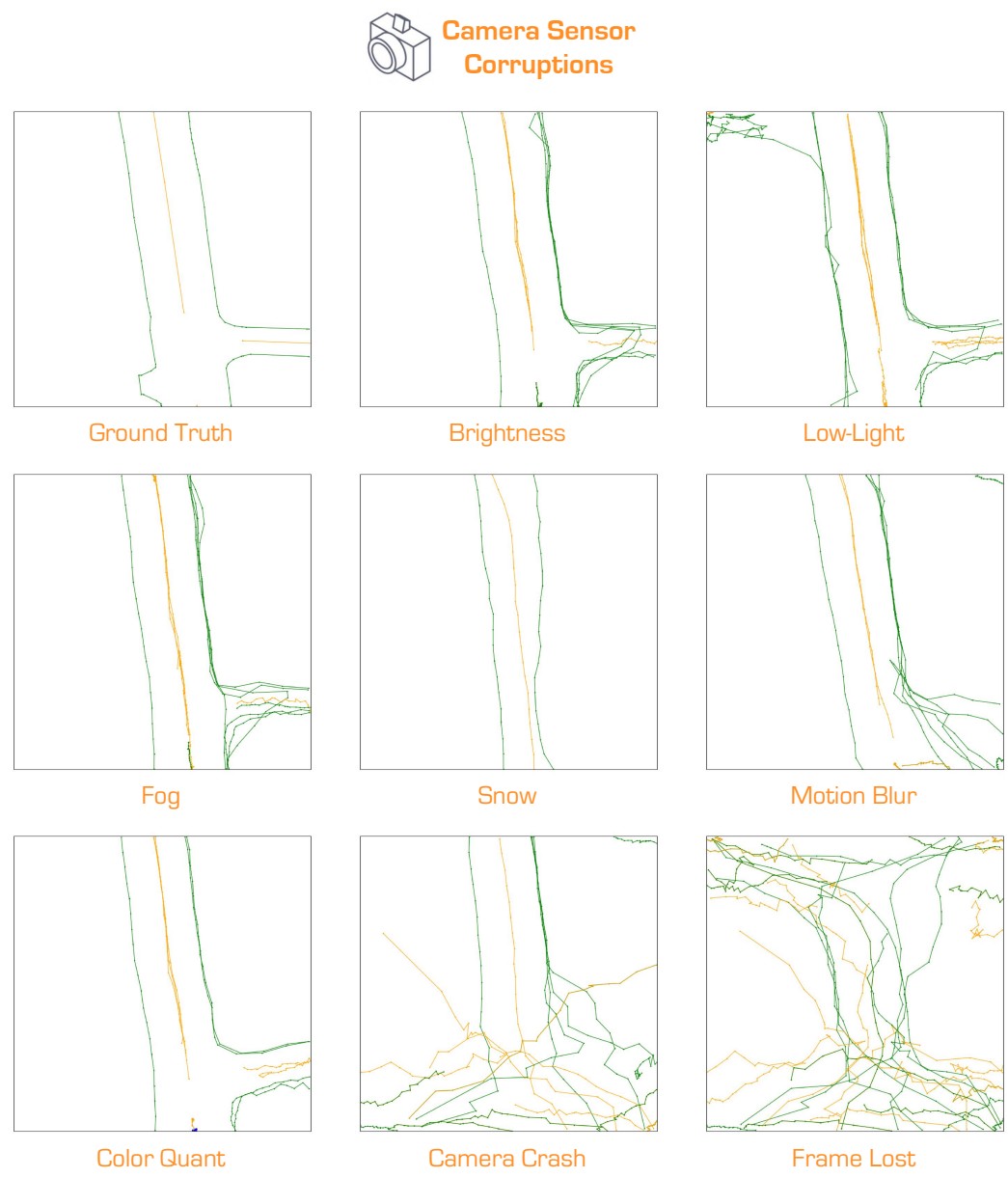

Figure 12: Qualitative assessment of camera-only HD map construction under the Camera sensor corruptions. Best viewed in color and zoomed in for details.

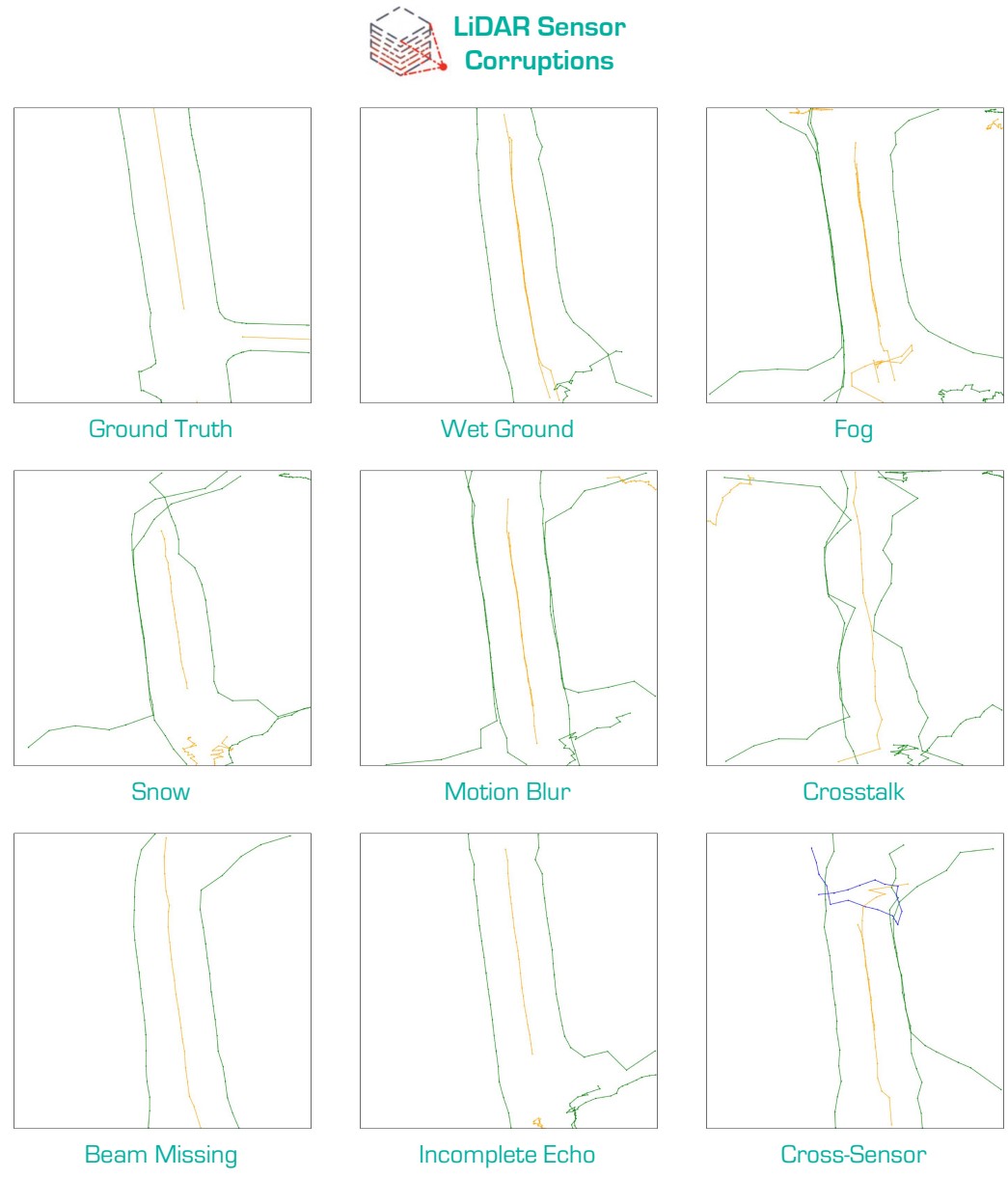

Figure 13: Qualitative assessment of LiDAR-only HD map construction under the LiDAR sensor corruptions. Best viewed in color and zoomed in for details.

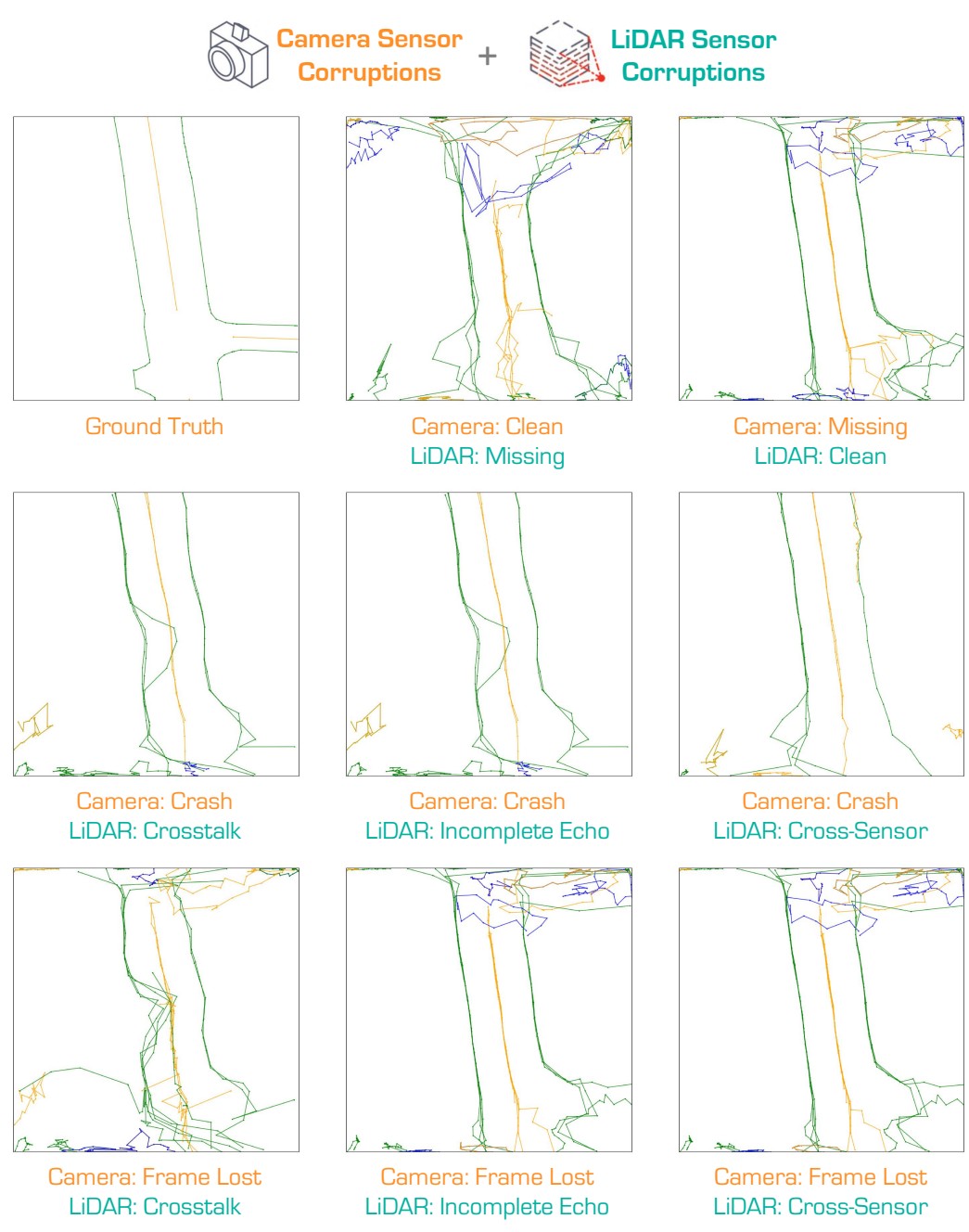

Figure 14: Qualitative assessment of camera-LiDAR fusion-based HD map construction under the Camera and LiDAR combined sensor corruptions. Best viewed in color and zoomed in for details.

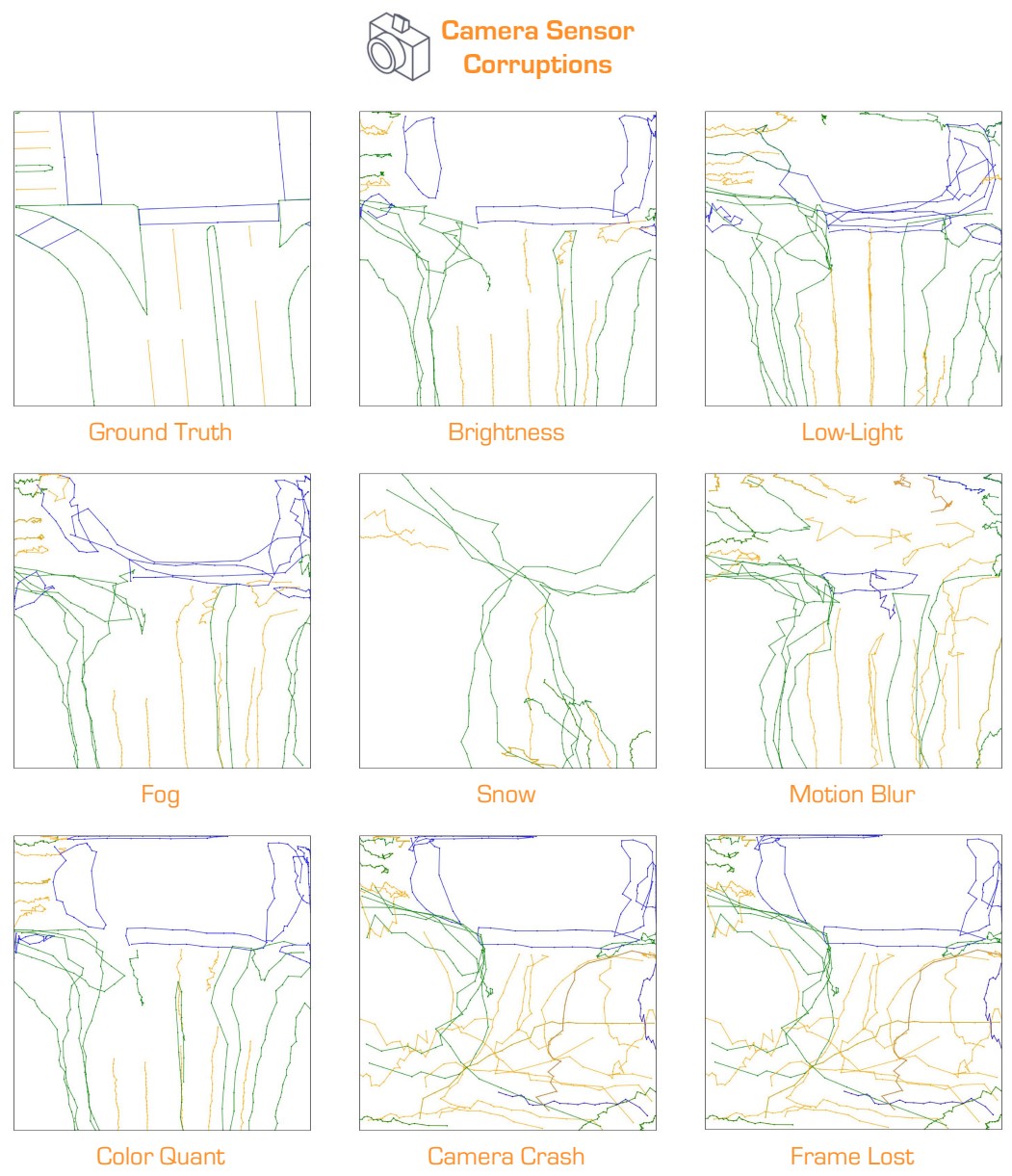

Figure 15: Qualitative assessment of camera-only HD map construction under the Camera sensor corruptions. Best viewed in color and zoomed in for details.

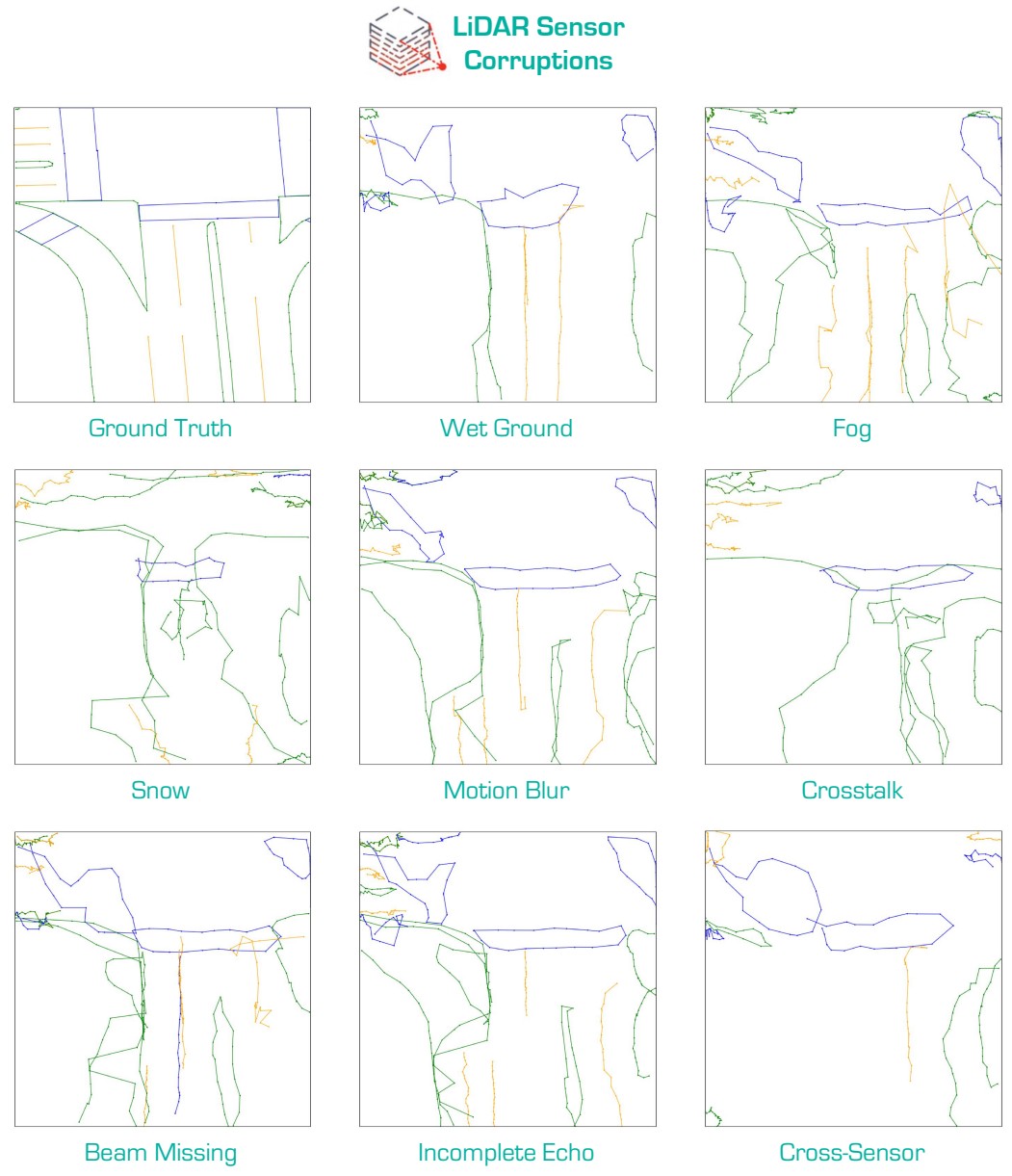

Figure 16: Qualitative assessment of LiDAR-only HD map construction under the LiDAR sensor corruptions. Best viewed in color and zoomed in for details.

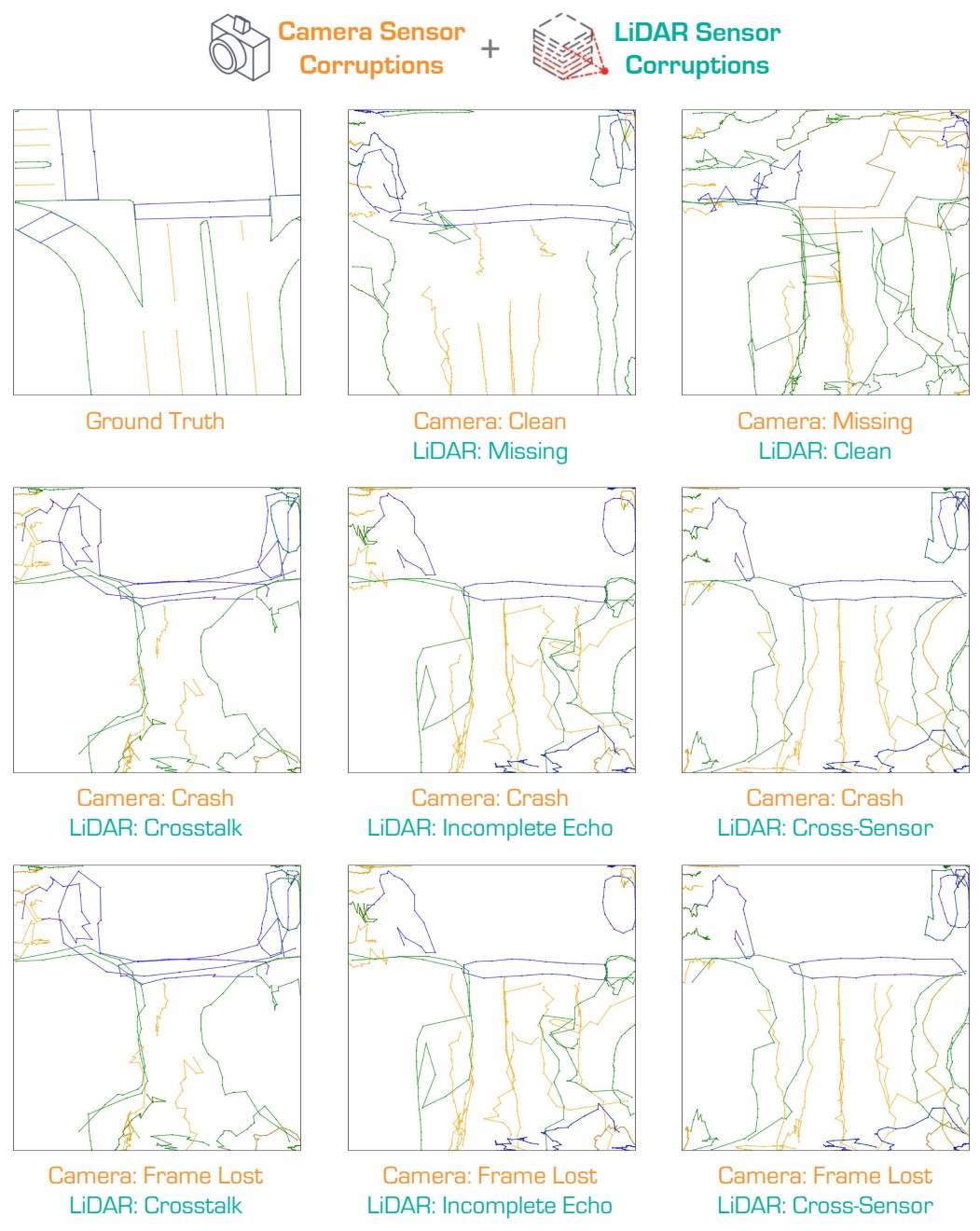

Figure 17: Qualitative assessment of camera-LiDAR fusion-based HD map construction under the Camera and LiDAR combined sensor corruptions. Best viewed in color and zoomed in for details.

# I Public Resources Used

In this section, we acknowledge the use of the public datasets, models, and codebase, during the course of this work.

## I.1 Public Datasets Used

We acknowledge the use of the following public datasets, during the course of this work:

- nuScenes[2] ................................................................ CC BY-NC-SA 4.0
- nuScenes-devkit[3] ............................................................ Apache License 2.0

## I.2 Public Models Used

We acknowledge the use of the following public implementations, during the course of this work:

- HDMapNet[4] .................................................................. GPL-3.0 license
- VectorMapNet[5] ............................................................... GPL-3.0 license
- PivotNet[6] ...................................................................... MIT license
- BeMapNet[7] .................................................................. Apache License 2.0
- MapTR[8] ........................................................................ MIT license
- MapTRv2[9] ..................................................................... MIT license
- StreamMapNet[10] ............................................................. GPL-3.0 license
- HIMap[11] ........................................................................ MIT license

## I.3 Public Codebase Used

We acknowledge the use of the following public codebases, during the course of this work:

- mmdetection3d[12] .......................................................... Apache License 2.0
- Robo3D[13] ................................................................... CC BY-NC-SA 4.0
- RoboDepth[14] ............................................................... CC BY-NC-SA 4.0

---

[2] https://www.nuscenes.org/nuscenes.

[3] https://github.com/nutonomy/nuscenes-devkit.

[4] https://github.com/Tsinghua-MARS-Lab/HDMapNet.

[5] https://github.com/Tsinghua-MARS-Lab/vectormapnet.

[6] https://github.com/wenjie710/PivotNet.

[7] https://github.com/er-muyue/BeMapNet.

[8] https://github.com/hustvl/MapTR.

[9] https://github.com/hustvl/MapTR.

[10] https://github.com/yuantianyuan01/StreamMapNet.

[11] https://github.com/BritaryZhou/HIMap.

[12] https://github.com/open-mmlab/mmdetection3d.

[13] https://github.com/ldkong1205/Robo3D.

[14] https://github.com/ldkong1205/RoboDepth.

