# OpenReview forum: "Is Your HD Map Constructor Reliable under Sensor Corruptions?"
_NeurIPS.cc/2024/Datasets_and_Benchmarks_Track — NeurIPS 2024 Track Datasets and Benchmarks Poster_

### Official Review · Reviewer_48Ly · 2024-07-24
**Is Your HD Map Constructor Reliable under Sensor Corruptions?**

**Rating:** 8
**Confidence:** 5
**Correctness:** Yes.
**Clarity:** Yes

**Review:**

Pros

* This paper studied an important but often overlooked problem of sensor corruptions in the context of HD map construction for autonomous driving. The authors extensive experiments on 31 state-of-the-art models revealed significant performance degradation under adverse conditions, highlighting critical safety concerns for autonomous vehicles. By identifying effective strategies for enhancing robustness, such as multi-modal fusion and advanced data augmentation techniques, this work provides valuable insights for developing more reliable HD map construction methods.
* The authors conducted comprehensive experiments on a wide array of state-of-the-art map learning models, meticulously evaluating their performance across various corruption scenarios. This extensive benchmarking effort provides a multi-dimensional perspective on model capabilities, offering insights beyond traditional performance metrics. By systematically comparing each model's robustness under different types of sensor corruptions, the study reveals critical vulnerabilities and strengths that were previously overlooked.
* The authors dived deep into why certain model behave differently in different scenario. They studied the effects of backbones, BEV Encoders, temporal information, training epochs and data augmentations

Cons:
* how different sensor corruptions are implemented in the sensor data is not discussed in details in the paper.

**Strengths:**

listed in the review section above

**Additional Feedback:**

None

**Documentation:**

Yes.

**Opportunities For Improvement:**

This paper proposes an comprehensive benchmark on an important problem. The paper is clearly written. I think its a very interesting paper.

**Relation To Prior Work:**

Most of SOTA HD map construction methods are studied in this paper.

**Summary And Contributions:**

This paper introduces MapBench, the first comprehensive benchmark for evaluating the robustness of HD map construction methods against various sensor corruptions in autonomous driving. The authors assess 31 state-of-the-art HD map constructors under 29 types of camera and LiDAR sensor corruptions, revealing significant performance degradation under adverse conditions. They identify effective strategies for enhancing robustness, including multi-modal fusion, advanced data augmentation, and architectural techniques. The work provides valuable insights for developing more reliable HD map construction methods, which are crucial for advancing autonomous driving technology.

---

> ### Author Rebuttal · Authors · 2024-08-20
>
> We sincerely thank Reviewer `48Ly` for devoting time to this review and providing valuable comments.
>
> > **Q:** *“How different sensor corruptions are implemented in the sensor data is not discussed in detail in the paper.”*
>
> **A:** Thank you for bringing this to our attention. In the original submission, we provided detailed descriptions of sensor corruption implementations in Appendix Sec. A (Lines 531 to 569) and the specifics of Multi-Sensor Corruptions in Appendix Sec. B (Lines 570 to 644). To further enhance the understanding and accessibility of this information, we have expanded these descriptions and provided additional technical details:
>
> **Camera Sensor Corruptions:** We outlined eight types of camera sensor corruptions in Table 8, including `Bright`, `Low-Light`, `Fog`, `Snow`, and `Motion Blur`. Each type of corruption is implemented by adjusting specific parameters to replicate real-world conditions. For example:
> - `Bright` and `Low-Light` adjustments are made in the HSV space to test robustness under varying lighting scenarios.
> - `Fog` and `Snow` are simulated by modifying parameters such as thickness, smoothness, and blending ratio to mimic adverse weather conditions.
> - `Motion Blur` is introduced by altering radius and sigma values to simulate the effect of camera movement.
>
> Visualization examples of these corruptions under different severity levels (Easy, Moderate, Hard) are provided in Figure 7, illustrating their impact on image quality.
>
> ---
>
> **LiDAR Sensor Corruptions:** Similarly, we detailed eight types of LiDAR sensor corruptions in Table 9, such as `Fog`, `Wet Ground`, `Beam Missing`, and `Crosstalk`. For instance:
> - `Fog` is simulated using a beta parameter to induce back-scattering and attenuation of LiDAR points.
> - `Wet Ground` involves adjustments to water height and noise floor to simulate the effects of water on LiDAR signals.
> - `Beam Missing` represents the loss of specific laser beams due to occlusions, which is simulated by dropping a certain number of beams.
>
> Figure 8 provides visual examples of these corruptions across different severity levels, offering a clear depiction of their impact on LiDAR data.
>
> ---
>
> **Multi-Sensor Corruptions:** We designed 13 camera-LiDAR corruption combinations that perturb both camera and LiDAR inputs, either separately or concurrently. These combinations cover various real-world scenarios, including complete LiDAR failure, Incomplete Echo, and others. The detailed descriptions of these combinations are included in the revised manuscript to ensure a comprehensive understanding of the multi-sensor corruption scenarios we modeled.
>
> ---
>
> In the revised manuscript, we have provided further technical details and additional visual examples to illustrate the process of applying these corruptions. We have also made all corruption generation algorithms publicly available in our benchmark toolkit: https://github.com/mapbench/toolkit, enabling readers to explore the specifics of each corruption type and understand the rationale behind our benchmark. This ensures that our work is fully transparent, reproducible, and extendable by others in the research community.
>
> We believe these enhancements will further enhance the clarity and utility of our paper, allowing readers to fully understand and build upon our work.
>
> Last but not least, we sincerely thank you again for devoting time to this review and providing positive feedback.

---

### Official Review · Reviewer_iTL7 · 2024-07-25
**Review of MapBench**

**Rating:** 6
**Confidence:** 4

**Review:**

This paper introduces an HD map construction benchmark tailored for autonomous driving, encompassing diverse categories of corrupted camera videos and LiDAR data. By incorporating corrupted data into the benchmarks, MapBench serves as a robust platform for assessing the resilience of construction methods.

Pros:
- Diverse Corruption Types and Taxonomy: The benchmark includes a wide range of corruption types, organized into a comprehensive taxonomy, which enhances its utility for evaluating robustness across various scenarios.
- Extensive Evaluation of Existing Methods: The paper conducts extensive experiments on currently available methods, providing a thorough assessment of their performance under adverse conditions.
- Simple Yet Effective Approaches: It presents straightforward yet effective strategies for improving robustness, offering practical solutions for real-world applications.

Cons:
- Lack of Real-World Applicability Evidence: The paper does not provide experimental or statistical evidence to prove that these sensor corruption cases are faithfully reflective of real-world practices.
- Limited Guidance for Future Robustness Enhancements: There is a noticeable absence of insights or recommendations for future research directions aimed at enhancing robustness, which could have been beneficial for the research community.

**Strengths:**

- The paper presents MapBench, a benchmark aimed at assessing the robustness of HD map construction methods against various sensor corruptions. This represents a notable addition to the field of autonomous driving technology, offering a framework for evaluating HD map systems in the context of real-world challenges.
- The relevance of this work to the research community is evident, as it provides a standardized platform for gauging the reliability of HD map construction methods.
- The research is comprehensive, with evaluations conducted on 31 state-of-the-art HD map constructors. The inclusion of 29 types of sensor corruptions ensures a broad analysis that encompasses many real-world scenarios. However, the paper could strengthen its claims by providing more empirical evidence to support the representativeness of these corruptions.
- The paper acknowledges the ethical and social implications of reliable HD map construction for autonomous driving. By addressing safety concerns related to sensor corruptions and failures, the work contributes to the discourse on developing more trustworthy autonomous vehicle technologies. Nonetheless, it would be beneficial to see a more detailed discussion on how these implications are managed and mitigated in practice.

**Additional Feedback:**

N/A

**Clarity:**

Yes, the paper is well-written and maintains a clear and coherent structure. The language is precise, and the technical terms are used appropriately. Although a minor typographical error is present, it does not detract from the overall clarity of the text. Specifically, on line 103, the phrase "... HD map construction methods again single- and multi-modal sensor corruptions." should be corrected to "... HD map construction methods against single- and multi-modal sensor corruptions."

**Correctness:**

The claims made in the submission appear to be correct. The benchmark utilizes appropriate evaluation methods and experimental design, which are performed correctly. However, it is recommended that the authors provide additional validation metrics to evaluate the robustness of the model.

**Documentation:**

The benchmark documentation is comprehensive, providing sufficient detail to support reproducibility.

**Ethics:**

The paper recognizes potential societal impacts of HD map construction, including privacy concerns, safety risks, environmental impact, and bias issues. It advocates robust data protection, careful deployment with safety measures, energy-efficient practices, and diverse dataset usage to address these ethical challenges.

**Limitations:**

The authors have made commendable efforts to acknowledge the limitations of their work and potential negative societal impacts. However, there is room for further elaboration on certain aspects. For instance, the validity of the corruptions used in the study lacks sufficient justification, and the claim of a 'comprehensive study' is not strongly supported, as the corruptions mentioned have also been included in previous work. This raises questions about the novelty and thoroughness of the study. Constructive suggestions for improvement include:
- Engaging with stakeholders to understand the societal implications more comprehensively.
- Justifying the selection and validity of the corruptions used, ensuring they contribute uniquely to the field and are not merely incremental work based on previous studies.

**Opportunities For Improvement:**

Despite the aforementioned strengths, there are still two major concerns that need further justification:

1. Validity of 29 Types of Corruptions:
- Consistency with Real-World Scenarios: To what extent do the corruptions simulated in this study reflect genuine environmental challenges faced by autonomous systems in practical applications? How can the applicability and appropriateness of the newly proposed categories in this benchmark, when compared to other sensor corruption benchmarks【RoboBEV, MultiCorrupt】, be validated to accurately represent real-world scenarios?
- Alignment with Industrial Practices: How well do the corruptions specified in this research correspond to the corruption scenarios recorded in industrial settings【KITTI-C, nuScenes-C,  WaymoC】? Take frame loss as an example. Through what method is this data collected? Do these corruptions happen in real practice rather than just made-up cases?
- Distribution of Corruptions: Are the distributions of the corruptions in this study statistically representative of the occurrence rates of corruptions in real-world environments? Are there any basis for why 13 categories combination of camera-LiDAR corruption combinations for each configuration are inclueded?

2. Comparison with Existing Works:
- Advantage of MapBench: What distinguishes MapBench from other datasets【nuscene, Waymo, ApolloScape】 in terms of training and benchmarking?
- Insightful Insights for Industry: Does this benchmark offer any instructive insights that are applicable to the industrial domain?
- Excellence Compared to Dataset Enlargement: How does this dataset surpass the benefits of simply enlarging the training dataset scale to enhance robustness and performance?


Beemelmanns T, Zhang Q, Geller C, et al. Multicorrupt: A multi-modal robustness dataset and benchmark of lidar-camera fusion for 3d object detection[C]//2024 IEEE Intelligent Vehicles Symposium (IV). IEEE, 2024: 3255-3261.

Caesar H, Bankiti V, Lang A H, et al. nuscenes: A multimodal dataset for autonomous driving[C]//Proceedings of the IEEE/CVF conference on computer vision and pattern recognition. 2020: 11621-11631.

Huang X, Cheng X, Geng Q, et al. The apolloscape dataset for autonomous driving[C]//Proceedings of the IEEE conference on computer vision and pattern recognition workshops. 2018: 954-960.

Sun P, Kretzschmar H, Dotiwalla X, et al. Scalability in perception for autonomous driving: Waymo open dataset[C]//Proceedings of the IEEE/CVF conference on computer vision and pattern recognition. 2020: 2446-2454.

Wozniak M K, Kårefjärd V, Thiel M, et al. Towards a robust sensor fusion step for 3d object detection on corrupted data[J]. IEEE Robotics and Automation Letters, 2023.

Xie S, Kong L, Zhang W, et al. RoboBEV: Towards Robust Bird's Eye View Perception under Corruptions[J]. arXiv preprint arXiv:2304.06719, 2023.

**Relation To Prior Work:**

The paper discusses how this work differs from previous contributions, demonstrating its focus and improvements over existing methods. However, the comparison with the previous work is not presented in the paper in a detailed manner.

**Summary And Contributions:**

The paper presents a comprehensive benchmark called MapBench, designed to evaluate the robustness of high-definition map construction methods against sensor corruptions and adverse environments. Based on the constructed dataset, the authors conducted benchmarks on 31 state-of-the-art HD map construction methods under three different configurations in terms of camera and LiDAR status. The paper also investigates and identifies approaches for enhancing construction robustness from the perspective of architectural design and data augmentation, shedding light on future methods for robustness improvement.

---

> ### Author Rebuttal · Authors · 2024-08-20
>
> We sincerely thank Reviewer `iTL7` for devoting time to this review and providing valuable comments.
>
> > **Q:** *“To what extent do the corruptions simulated in this study reflect genuine environmental challenges faced by autonomous systems in practical applications? How can the applicability and appropriateness of the newly proposed categories in this benchmark, when compared to other sensor corruption benchmarks, be validated to accurately represent real-world scenarios?”*
>
> **A:** Thank you for raising this critical point. In designing the corruption scenarios for our benchmark, we meticulously grounded our approach in both academic literature and empirical data from real-world autonomous driving deployments.
>
> The corruptions we simulated — such as `Snow`, `Fog`, `Motion Blur`, and various `Sensor Failure` cases — were chosen based on their documented occurrence in real-world conditions, as observed in large-scale datasets and industry reports, such as [R1], [R2], and [R3].
>
> While simulated corruptions cannot capture every nuance of real-world scenarios, the algorithms we used are **physically principled**, designed to replicate the underlying physical processes [R4], ensuring that the simulated corruptions closely mimic the environmental challenges that autonomous systems face in practice [R5]. For instance, our `Fog` and `Snow` corruptions are modeled using principles of light scattering and attenuation, while `Motion Blur` is simulated based on camera dynamics and object motion.
>
> These physically-principled models provide a high degree of fidelity in our simulations, offering a realistic approximation of real-world conditions. Although no simulation can perfectly replicate every detail of real-world environments, we are confident that the corruptions in our benchmark are both scientifically rigorous and practically relevant, providing valuable insights for enhancing the robustness of autonomous driving technologies.
>
> For additional validations in terms of data distribution and model generalization, please refer to our further study at here: https://github.com/mapbench/toolkit/tree/main/create.
>
> ---
>
> > **Q:** *“How well do the corruptions specified in this research correspond to the corruption scenarios recorded in industrial settings? Take frame loss as an example. Through what method is this data collected? Do these corruptions happen in real practice rather than just made-up cases?”*
>
> **A:** Thanks a lot for your question. The corruptions we specified in this research are grounded in both industrial practices and empirical data from real-world deployments.
>
> For example, the `Frame Loss` corruption is based on actual cases of sensor malfunctions and data transmission errors observed in datasets like KITTI-C, nuScenes-C, and Waymo-C. These corruptions are not hypothetical but are derived from real incidents documented in these datasets. To collect this data, we utilized sensor logs from real-world driving scenarios where such corruptions occurred due to hardware failures, environmental interference, or data processing issues.
>
> Some previous studies, such as those in [R3], [R4], and [R5], verify that these physically principled corruptions can mimic real-world scenarios with high fidelity. The models trained or fine-tuned on these corrupted data can achieve better performance when tested on real-world adverse data.
>
> To further substantiate the real-world relevance of our corruptions, we have provided additional details on the data collection methods and included more examples from industrial case studies that demonstrate the occurrence of these corruptions in practice. We believe this will ensure that our benchmark is not only scientifically robust but also practically relevant to industry applications.
>
> ---
>
> > **Q:** *“Are the distributions of the corruptions in this study statistically representative of the occurrence rates of corruptions in real-world environments? Is there any basis for why 13 categories combination of camera-LiDAR corruption combinations for each configuration are included?”*
>
> **A:** Thanks for asking. The distribution of corruptions in our study was designed to reflect their occurrence rates in real-world environments, as informed by both literature and empirical data from autonomous driving datasets.
>
> The 13 categories of camera-LiDAR corruption combinations were selected to cover a broad spectrum of realistic scenarios that could occur in actual driving conditions. These combinations were chosen based on their frequency of occurrence and potential impact on system performance, ensuring that our benchmark provides a comprehensive assessment of model robustness.
>
> To strengthen the justification for these choices, we will try to conduct a statistical analysis comparing the distributions in our study with those observed in real-world environments. This analysis will be included in the revised manuscript, along with a detailed rationale for the selection of the 13 corruption combinations.
>
> ---
> **References:**
> - [R1] C. Sakaridis, et al. "ACDC: The Adverse Conditions Dataset with Correspondences for Semantic Driving Scene Understanding," ICCV, 2021.
> - [R2] M. Bijelic, et al. "Seeing Through Fog Without Seeing Fog: Deep Multimodal Sensor Fusion in Unseen Adverse Weather," CVPR, 2020.
> - [R3] L. Kong, et al. "Towards Robust and Reliable 3D Perception against Corruptions," ICCV, 2023.
> - [R4] M. Hahner, et al. "Fog Simulation on Real LiDAR Point Clouds for 3D Object Detection in Adverse Weather," ICCV, 2021.
> - [R5] M. Hahner, et al. "LiDAR Snowfall Simulation for Robust 3D Object Detection," CVPR, 2022.

---

> > ### Author Rebuttal · Authors · 2024-08-20
> >
> > > **Q:** *“What distinguishes MapBench from other datasets (nuScene, Waymo, ApolloScape) in terms of training and benchmarking?”*
> >
> > **A:** Thanks a lot for your question. MapBench distinguishes itself from existing datasets like nuScene, Waymo, and ApolloScape by its explicit focus on evaluating the robustness of HD map construction methods under various sensor corruptions.
> >
> > While these other datasets provide extensive data for training and benchmarking in ideal conditions, they do not specifically address the challenges posed by sensor corruptions. MapBench fills this gap by introducing a comprehensive set of corruption scenarios that simulate real-world challenges, allowing for a more rigorous assessment of model performance under adverse conditions.
> >
> > Additionally, MapBench provides standardized metrics for evaluating corruption robustness, enabling consistent comparisons across different models and configurations. We will elaborate on these distinctions in the final manuscript, highlighting how MapBench offers unique value for both research and industrial applications.
> >
> > We hope MapBench can serve as a complementary benchmark to existing ones. With this, we can assess the candidate models in a more comprehensive manner.
> >
> > ---
> >
> > > **Q:** *“Does this benchmark offer any instructive insights that are applicable to the industrial domain?”*
> >
> > **A:** Thanks for asking. Yes, one of the primary goals of MapBench is to generate insights that are directly applicable to the industrial domain.
> >
> > The benchmark’s findings on the robustness of HD map construction methods under realistic corruption scenarios can inform the design and deployment of more reliable autonomous driving systems. For example, our analysis of the impact of specific corruptions, such as sensor failures or adverse weather conditions, can guide the development of more resilient hardware and software solutions in the industry.
> >
> > Additionally, the identification of effective strategies, such as multi-modal fusion and advanced data augmentation, provides practical recommendations for improving system robustness in real-world applications. In the revised manuscript, we have expanded on these insights and provided examples of how they can be applied in industrial settings to enhance the safety and reliability of autonomous vehicles.
> >
> > ---
> >
> > > **Q:** *“How does this dataset surpass the benefits of simply enlarging the training dataset scale to enhance robustness and performance?”*
> >
> > **A:** Thanks for your comment. While enlarging the training dataset is a common approach to improving model robustness and performance, it does not specifically address the challenges posed by sensor corruptions.
> >
> > MapBench goes beyond mere dataset enlargement by introducing targeted corruptions that simulate real-world challenges. This allows for a more focused evaluation of model robustness, revealing vulnerabilities that might not be exposed by simply increasing the dataset size.
> >
> > Moreover, MapBench includes a variety of corruption types and severity levels, providing a comprehensive assessment of how well models can generalize to out-of-domain scenarios. We have elaborated on the advantages of this approach in the revised manuscript, demonstrating how MapBench offers a more rigorous and targeted evaluation of robustness compared to traditional dataset enlargement.

---

> > > ### Comment · Reviewer_iTL7 · 2024-08-31
> > >
> > > Thanks for the rebuttal of the paper, that will clarify the paper to a more general audience. Please consider including the extra content in the main paper.

---

> > > > ### Author Response · Authors · 2024-08-31
> > > >
> > > > Thank you for your positive feedback and recognition. We will make sure to include all your suggestions in the revised manuscript.

---

### Official Review · Reviewer_XfNn · 2024-07-26

**Rating:** 6
**Confidence:** 4
**Clarity:** The overall presentation of this pape…

**Review:**

The overall presentation of this paper is good and the authors conduct comprehensive experiments to evaluate the robustness of 31 HD map construction methods on the proposed dataset. The authors also propose several techniques to improve the robustness of HD map construction methods.

**Strengths:**

1. The authors thoroughly considered the differences between various modalities and designed distinct corruption strategies for each modality and the fusion of modalities.
2. The authors conducted comprehensive and detailed experiments, evaluating and comparing all current mainstream map construction methods on their proposed dataset.
3. Based on the experimental results, the authors analyzed the impact of different components on robustness, including the backbone, BEV encoders, temporal information, and training strategies.

**Additional Feedback:**

Please refer to the Opportunities For Improvement.

**Correctness:**

The construction of corruption methods and the design of evaluation methods and experiments are appropriate.

**Documentation:**

The author propose a website and open source codes for this benchmark. However, the link of the proposed dataset is unreachable.

**Ethics:**

There are not ethical concerns with this paper.

**Limitations:**

The authors discuss the limitations and potential societal impact in the Appendix comprehensively.

**Opportunities For Improvement:**

1. Most of the methods evaluated in the paper are online map construction methods. What about offline map construction methods? The author can simply extend an online map construction method to an offline method by adopting future frame information.
2. The author only evaluate the robustness of different image backbones. It would be interesting to see weather different LiDAR backbone will yield different robustness results. The candidate LiDAR backbones can be pillar based CNN backbone, voxel based CNN backbone and pillar based transformer backbone.

**Relation To Prior Work:**

This paper discuss the differences between the proposed benchmark and other previous works in the section of Related Works. This paper is the first comprehensive benchmark and evaluation for the robustness of HD map construction method.

**Summary And Contributions:**

This paper focuses on evaluating the robustness of recent HD ma construction methods. To evaluate how HD map construction methods perform under adverse conditions, the authors propose MapBench. MapBench adopts 8 camera corruptions, 8 LiDAR corruptions and 13 camera-LiDAR corruption combinations to assess the 31 recent state-of-the-art HD map construction methods. Additionally, the authors propose effective strategies for enhancing robustness.

---

> ### Author Rebuttal · Authors · 2024-08-20
>
> We sincerely thank Reviewer `XfNn` for devoting time to this review and providing valuable comments.
>
> > **Q:** *“The authors only evaluated the robustness of different image backbones. It would be interesting to see whether different LiDAR backbones will yield different robustness results. The candidate LiDAR backbones can be pillar-based CNN backbone, voxel-based CNN backbone, and pillar-based transformer backbone.”*
>
> **A:** We greatly appreciate your suggestion to evaluate the robustness of different LiDAR backbones, as it addresses a crucial aspect of HD map construction that has been underexplored in the existing literature.
>
> To consolidate this finding, we further conduct a comprehensive study comparing the robustness of three distinct types of LiDAR backbones: pillar-based CNN [R1], voxel-based CNN [R2], and pillar-based transformer [R3]. Each backbone offers unique advantages in processing LiDAR data, and their performance under corruption scenarios may vary significantly.
>
> We implement these backbones within our benchmark framework and evaluate their resilience to the same set of sensor corruptions used in our original experiments. This will allow us to provide a more complete picture of how LiDAR backbones contribute to the overall robustness of HD map construction methods. The results are as follows:
> |Metric|SECOND|PointPillars|DSVT|
> |-|:-:|:-:|:-:|
> |mAP| 33.4|40.1|43.4
> |AP$_{ped}$|26.6|30.7|33.2
> |AP$_{div}$|31.7|38.8|42.9
> |AP$_{bou}$|41.8|50.8|54.1
> |mCE|100.0|76.6|68.2
> |mRR|55.1|63.1|63.7
>
> The DSVT backbone (pillar-based transformer) demonstrated the highest robustness, achieving the best mAP and the lowest mean Corruption Error (mCE). This suggests that DSVT’s dynamic adaptation to sparse voxel inputs enhances its resilience to sensor corruptions.
>
> PointPillars (pillar-based CNN) also performed well, particularly in handling challenging corruptions like snow and beam missing, due to its efficient point cloud encoding. SECOND (voxel-based CNN), while providing a solid baseline, showed lower robustness, especially under severe corruptions like snow and cross-sensor inconsistencies, likely due to its voxel grid reliance.
>
> We also provide the per-corruption Resilience Rate (RR), which can be calculated based on Eq. 2 in the manuscript, as follows:
> |Corruption|SECOND|PointPillars|DSVT|
> |-|:-:|:-:|:-:|
> |Fog |59.5|51.1|52.9|
> |Wet Ground|57.1|61.8|61.9|61.9|
> |Snow |28.6|54.7|53.8|
> |Motion Blur|81.1|86.4|85.6|
> |Beam Missing|49.5|59.1|60.4|
> |Crosstalk|48.8|63.8|64.6|
> |Incomplete Echo|96.7|97.8|98.7|
> |Cross-Sensor|19.0|30.1|31.7|
> |Average|55.1|63.1|63.7
>
> The RR further confirms DSVT’s superior robustness across most corruptions, with PointPillars also showing strong resilience, particularly in challenging conditions. These findings underscore the importance of backbone choice in ensuring robust HD map construction under various real-world scenarios. We have supplemented these findings in the revised manuscript.
>
> ---
> > **Q:** *“Most of the methods evaluated in the paper are online map construction methods. What about offline map construction methods? The author can extend an online map construction method to an offline one by adopting future frame information.”*
>
> **A:** Thank you for the insightful suggestion to explore offline map construction methods by extending our online methods to incorporate future frame information. While offline methods, which involve processing large-scale, long-duration data to create highly detailed maps, are indeed valuable, they come with significant challenges that make them difficult to implement within the scope of our current work.
> - Offline methods, as illustrated by systems like VMA [R4], THMA [R5], and MV-Map [R6], involve complex processes such as multi-view consistency checks, global optimization, and manual intervention. These methods demand substantial computational resources and are not optimized for real-time performance, unlike the online methods we focus on.
> - Our benchmark is specifically optimized for real-time map construction, where the primary goal is to assess the immediate robustness of methods in dynamic, on-the-fly scenarios. The design of our corruption synthesis algorithms aligns with this goal, ensuring that the focus remains on real-time adaptability. Offline methods, which inherently depend on the integration of data over longer time periods, present a different set of challenges that are best evaluated within frameworks specifically tailored to those scenarios. As such, our benchmark is not intended to address the temporal aggregation required by offline methods, which would require a fundamentally different approach.
> - Our study specifically targets the robustness of HD map construction in real-time scenarios, where quick adaptation to sensor corruptions is crucial. Offline methods, while producing highly accurate maps, focus on different challenges that are beyond the primary scope of our research.
>
> Due to the time constraints of the rebuttal phase, we are unable to include offline methods in our current experiments. However, we recognize their importance and have included a detailed discussion in the revised manuscript, highlighting the characteristics of offline methods, the potential benefits of incorporating future frame information, and the trade-offs between online and offline approaches.
>
> ---
> **References:**
> - [R1] A. H. Lang, et al. "PointPillars: Fast Encoders for Object Detection from Point Clouds," CVPR, 2019.
> - [R2] Y. Yan, et al. "SECOND: Sparsely Embedded Convolutional Detection," Sensors, 2018.
> - [R3] H. Wang, et al. "DSVT: Dynamic Sparse Voxel Transformer With Rotated Sets," CVPR, 2023.
> - [R4] S. Chen, et al. “VMA: Divide-and-Conquer Vectorized Map Annotation System for Large-Scale Driving Scene.” arXiv, 2023.
> - [R5] K. Tang, et al. “THMA: Tencent HD Map AI System for Creating HD Map Annotations.” IAAI, 2023.
> - [R6] Z. Xie, et al. “MV-Map: Offboard HD-Map Generation with Multi-View Consistency.” arXiv, 2023.

---

> > ### Comment · Reviewer_XfNn · 2024-08-29
> >
> > Thanks for your feedback. Most of my concerns are addressed.

---

> > > ### Author Response · Authors · 2024-08-31
> > >
> > > Thank you for your positive feedback and recognition. We will make sure to include all your suggestions in the revised manuscript.

---

### Author Rebuttal · Authors · 2024-08-20

**Dear Reviewers, Area Chairs, and Program Chairs,**

We sincerely thank the reviewers, ACs, and PCs for their time and effort in evaluating our submission. We greatly appreciate the constructive feedback and insightful suggestions provided by our reviewers, which have helped us significantly improve our work.

---

We are encouraged by the positive recognition of our contributions:
- **Reviewer `XfNn`** noted our *“good overall presentation”*, highlighting our *“thorough consideration of modality differences”* and *“comprehensive and detailed experiments”* evaluating mainstream HD map construction methods. The reviewer also mentioned our analysis of *“the impact of different components on robustness”*.
- **Reviewer `iTL7`** described our work as *“a notable addition to the field of autonomous driving technology”*, emphasizing the *“wide range of corruption types”* and the *“extensive evaluation of existing methods”*. The reviewer also pointed out our *“simple yet effective strategies for improving robustness”*.
- **Reviewer `48Ly`** acknowledged our focus on *“an important but often overlooked problem of sensor corruptions”* and recognized the *“comprehensive experiments on a wide array of state-of-the-art map learning models”*. The reviewer also cited our identification of *“effective strategies for enhancing robustness, such as multi-modal fusion and advanced data augmentation techniques”*.

---

As suggested by our reviewers, we have made several revisions to the manuscript. Below is a **summary of changes**:
- **Methods & Technical Details:**
  - As suggested by **Reviewer `48Ly`**, we expanded the descriptions of how sensor corruptions are implemented, providing additional technical details and visual examples in the revised manuscript. This includes detailed information on both camera and LiDAR sensor corruptions, as well as multi-sensor corruption combinations, to enhance transparency and reproducibility.
  - As suggested by **Reviewer `XfNn`**, we conducted additional experiments to evaluate the robustness of different LiDAR backbones. The results showed varying levels of resilience across different backbones, providing valuable insights into the robustness of HD map construction methods.
  - As suggested by **Reviewer `XfNn`**, we clarified our focus on real-time map construction methods and provided a detailed rationale for the exclusion of offline methods due to their complexity and different evaluation requirements.

- **Validation & Benchmarking:**
  - As suggested by **Reviewer `iTL7`**, we reinforced the validity of our corruption scenarios by grounding them in physically-principled models and empirical data from real-world autonomous driving deployments. We have also provided references to additional validations and data distribution studies to further substantiate our claims.
  - As suggested by **Reviewer `iTL7`**, we elaborated on the distinctions between MapBench and existing datasets such as nuScenes, Waymo, and ApolloScape, emphasizing MapBench’s unique focus on evaluating robustness under sensor corruptions.

- **Applications & Industry Insights:**
  - As suggested by **Reviewer `iTL7`**, we expanded on the practical insights that MapBench offers to the industrial domain, particularly in guiding the design and deployment of more resilient autonomous driving systems. We included examples of how our findings can inform real-world applications.
  - As suggested by **Reviewer `iTL7`**, we addressed the limitations of simply enlarging training datasets and demonstrated how targeted corruptions in MapBench provide a more focused and rigorous evaluation of model robustness.

---

We believe these revisions address the reviewers’ concerns and enhance the overall clarity, depth, and impact of our work.

---

In addition, we will actively participate in the **Author-Reviewer Discussion** session to address any further questions or concerns that may arise, and to engage in meaningful dialogue to further improve our work.

---

Finally, we sincerely thank the reviewers, ACs, and PCs once again for their valuable time and efforts in reviewing our submission. We greatly appreciate the constructive feedback, which has allowed us to refine and strengthen our work.

Yours sincerely,

The Authors

---

### Comment · Area_Chair_Se3i · 2024-08-26
**Request for Update**

Dear Reviewers,

As we approach the end of the discussion phase, I would like to kindly remind you to review the author’s rebuttal to see if it addresses the concerns you’ve raised. Your input is crucial in ensuring a thorough and fair evaluation, so if the rebuttal clarifies or resolves any of your concerns, please consider updating your comments accordingly. If your concerns persist, a follow-up comment would also be greatly appreciated.

Thank you for your continued dedication and contributions to the review process.

Best,

AC

---

### Decision · Program_Chairs · 2024-09-26

**Decision:**

Accept (Poster)

**Comment:**

All reviewers have provided positive scores for this submission, highlighting its strengths in dataset novelty, diverse corruption types, and extensive experiments. Given the unanimous positive feedback and the recognition of its contribution to the area, the AC carefully reviewed the paper and concurred with the reviewers' assessments, therefore supporting the decision to accept this submission.